# Modulation of brain cation-Cl− cotransport via the SPAK kinase inhibitor ZT-1a

Jinwei Zhang [1,2,14]*, Mohammad Iqbal H. Bhuiyan [3,14], Ting Zhang[4,14], Jason K. Karimy[5], Zhijuan Wu [6], Victoria M. Fiesler[3], Jingfang Zhang[4], Huachen Huang[3], Md Nabiul Hasan[3], Anna E. Skrzypiec[1], Mariusz Mucha[1], Daniel Duran [5], Wei Huang[4], Robert Pawlak[1], Lesley M. Foley[7], T. Kevin Hitchens[7,8], Margaret B. Minnigh[9], Samuel M. Poloyac[9], Seth L. Alper[10], Bradley J. Molyneaux[3,11], Andrew J. Trevelyan[12], Kristopher T. Kahle[5]*, Dandan Sun[3,13]* & Xianming Deng[4]*

The *SLC12A* cation-Cl− cotransporters (CCC), including NKCC1 and the KCCs, are important determinants of brain ionic homeostasis. SPAK kinase (*STK39*) is the CCC master regulator, which stimulates NKCC1 ionic influx and inhibits KCC-mediated efflux via phosphorylation at conserved, shared motifs. Upregulation of SPAK-dependent CCC phosphorylation has been implicated in several neurological diseases. Using a scaffold-hybrid strategy, we develop a novel potent and selective SPAK inhibitor, 5-chloro-N-(5-chloro-4-((4-chlorophenyl)(cyano)methyl)-2-methylphenyl)-2-hydroxybenzamide ("ZT-1a"). ZT-1a inhibits NKCC1 and stimulates KCCs by decreasing their SPAK-dependent phosphorylation. Intracerebroventricular delivery of ZT-1a decreases inflammation-induced CCC phosphorylation in the choroid plexus and reduces cerebrospinal fluid (CSF) hypersecretion in a model of post-hemorrhagic hydrocephalus. Systemically administered ZT-1a reduces ischemia-induced CCC phosphorylation, attenuates cerebral edema, protects against brain damage, and improves outcomes in a model of stroke. These results suggest ZT-1a or related compounds may be effective CCC modulators with therapeutic potential for brain disorders associated with impaired ionic homeostasis.

[1] Institute of Biomedical and Clinical Sciences, Medical School, College of Medicine and Health, University of Exeter, Hatherly Laboratories, Exeter EX4 4PS, UK. [2] Xiamen Cardiovascular Hospital, School of Medicine, Xiamen University, Xiamen, Fujian 361004, China. [3] Department of Neurology and Pittsburgh Institute For Neurodegenerative Diseases, University of Pittsburgh, Pittsburgh, PA 15213, USA. [4] State Key Laboratory of Cellular Stress Biology, Innovation Center for Cell Signaling Network, School of Life Sciences, Xiamen University, Xiamen, Fujian 361102, China. [5] Departments of Neurosurgery, Pediatrics, and Cellular & Molecular Physiology; Interdepartmental Neuroscience Program; and Centers for Mendelian Genomics, Yale School of Medicine, New Haven, CT 06511, USA. [6] Newcastle University Business School, Newcastle University, Newcastle upon Tyne, NE1 4SE, UK. [7] Animal Imaging Center, University of Pittsburgh, Pittsburgh, PA 15203, USA. [8] Department of Neurobiology, University of Pittsburgh, Pittsburgh, PA 15213, USA. [9] Department of Pharmaceutical Sciences, School of Pharmacy, University of Pittsburgh, Pittsburgh, PA 15261, USA. [10] Division of Nephrology, Department of Medicine, Beth Israel Deaconess Medical Center and Harvard Medical School, Boston, MA 02215, USA. [11] Department of Critical Care Medicine, University of Pittsburgh, Pittsburgh, PA 15213, USA. [12] Institute of Neuroscience, Medical School, Newcastle University, Framlington Place, Newcastle upon Tyne, NE2 4HH, UK. [13] Veterans Affairs Pittsburgh Health Care System, Geriatric Research, Educational and Clinical Center, Pittsburgh, PA 15213, USA. [14]These authors contributed equally: Jinwei Zhang, Mohammad Iqbal H. Bhuiyan, Ting Zhang. *email: j.zhang5@exeter.ac.uk; kristopher.kahle@yale.edu; sund@upmc.edu; xmdeng@xmu.edu.cn

Regulation of cellular ion transport is critical for brain water homeostasis. Vectorial ion transport across apical and basolateral membranes of the choroid plexus epithelium (CPe), accompanied by transport of water, cotransported[1–3] or osmotically obligated[4–7], results in daily cerebrospinal fluid (CSF) secretion of >500 cc/day into brain ventricular spaces. Impaired ionic homeostasis in CPe can result in hydrocephalus (accumulation of excess CSF in the brain ventricles), as in the settings of intraventricular hemorrhage (IVH) and infection[1–3]. Coordinated transmembrane influx and efflux of ions and water is also necessary for cell volume homeostasis in neurons, glia, and blood–brain barrier (BBB) endothelial cells. Impaired cell volume regulation following ischemic stroke and other brain injuries can lead to cytotoxic cell swelling, disruption of BBB integrity, and cerebral edema[8–10]. Hydrocephalus requires neurosurgical treatment by permanent, catheter-based CSF shunting, whereas treatment of ischemic stroke may require decompressive hemicraniectomy[11]. Both of these often morbid procedures have been used for decades with minimal innovation[12–14]. Recent advances in vascular stroke therapy such as clot lysis by recombinant tissue plasminogen activator (rtPA) and clot removal by radiologically guided thrombectomy are appropriate for fewer than 8% of ischemic stroke patients[15,16]. Thus, development of novel pharmacological modulators of brain ion transport is warranted to provide nonsurgical alternatives to current morbid treatments of these neurological disorders with altered volume homeostasis.

The electroneutral cation–Cl$^-$ cotransporters (CCCs) are secondary-active plasmalemmal ion transporters that utilize electrochemically favorable transmembrane gradients of Na$^+$ and/or K$^+$, established by the ouabain-sensitive Na$^+$, K$^+$-ATPase to drive transport of Cl$^-$ (and Na$^+$/K$^+$) into or out of cells. In epithelial cells under most physiological conditions (with the possible exception of choroid plexus), the Na$^+$-driven CCCs NCC, NKCC1, and NKCC2 ("N(KCCs)")[17–22], function as Cl$^-$ importers, whereas the Na$^+$-independent KCC1–4 ("KCCs")[23–26] function as Cl$^-$ exporters. These evolutionarily conserved transporters[27] are of particular importance in regulation of ion and water homeostasis in mammalian central nervous system (CNS)[28,29]. The coordinated regulation of CCC function is important for cell volume regulation in most brain cells, preventing excessive cell swelling or shrinkage in response to osmotic or ischemic stress[26,30–34]. The central importance of CCCs to choroid plexus regulation of CSF homeostasis has been recently recognized[1,3].

The Ste20-type Ser–Thr protein kinases SPAK (SPS1-related proline/alanine-rich kinase) and OSR1 (oxidative stress-responsive kinase 1) are considered master regulators of the CCCs[35]. SPAK and OSR1 are activated by phosphorylation in the regulatory "T-loop" (SPAK Thr233 and OSR1 Thr185) by one of four WNK ["with no lysine" (K)] protein kinases[17,36]. The WNK and SPAK/OSR1 protein kinases drive Cl$^-$ influx by phosphorylation and *activation* of the Na$^+$-driven CCC members (NCC, NKCC1, and NKCC2)[18–20] while inhibiting Cl$^-$ efflux by phosphorylation and *inactivation* of KCC1–4[25,26]. This reciprocal regulation of the Na$^+$- and K$^+$-driven CCCs by WNK–SPAK/OSR1 ensures tight coordination of cellular Cl$^-$ influx and efflux[23,37], and is essential for regulation of cell volume and epithelial transport in multiple tissues[38].

SPAK-regulated, CCC-mediated ion transport has been implicated in the pathogenesis of multiple brain pathologies associated with impaired brain ion and water homeostasis. Experimental ischemic cerebral edema is associated with increased phosphorylation of the SPAK/OSR1 T-loop and of NKCC1 (Thr$^{203}$/Thr$^{207}$/Thr$^{212}$) in both neurons and oligodendrocytes[39], and in BBB endothelial cells[40,41]. Mouse germline SPAK knockout significantly reduces ischemia-induced NKCC1 phosphorylation, infarct volume, axonal demyelination, and cerebral edema following ischemic stroke[39,42]. Choroid plexus NKCC1 is an essential mediator of ion transport in the CSF hypersecretory response that drives development of post-hemorrhagic hydrocephalus[3]. The 3.5-fold increase in CSF secretion accompanying the hydrocephalus caused by experimental IVH is associated with upregulated phosphorylation of SPAK/OSR1–NKCC1 at the choroid plexus apical membrane[3]. Indeed, the choroid plexus is the site of the highest SPAK abundance among all epithelial tissues[22]. Knockdown of SPAK in the choroid plexus by intracerebroventricular siRNAs reduced CSF secretion rates and reversed post-IVH ventriculomegaly[3].

Here, we report our development of a "dual" CCC modulator (NKCC1 inhibitor/KCC activator), 5-chloro-N-(5-chloro-4-((4-chlorophenyl)(cyano)methyl)-2-methylphenyl)-2-hydroxybenzamide ("ZT-1a") that potently and selectively inhibits SPAK kinase. ZT-1a-mediated SPAK inhibition led to reduced cellular ion influx and to stimulated Cl$^-$-dependent K$^+$ efflux by simultaneous reduction of the activating phosphorylation of NKCC1 and the inhibitory phosphorylation of the KCCs. Intracerebroventricular delivery of ZT-1a prevented CSF hypersecretion in a model of post-hemorrhagic hydrocephalus by decreasing SPAK-mediated phosphorylation of CCCs in the choroid plexus. Systemic ZT-1a administration after experimental ischemic stroke attenuated cerebral infarction and edema and improved neurological outcomes by decreasing SPAK-mediated phosphorylation of CCCs in brain tissues. These results suggest that inhibition of SPAK kinase is an effective approach in modulating CCCs and has therapeutic potential for brain disorders of cell volume dysregulation.

## Results

**ZT-1a, a novel and potent non-ATP-competitive SPAK inhibitor.** To identify pharmacological modulators of SPAK kinase, we designed and synthesized a new focused chemical library derived from the previously identified SPAK inhibitors Closantel[43], Rafoxanide[44], and STOCK1S-14279[43] (Fig. 1). This "scaffold-hybrid" strategy, combining pharmacophores from different scaffolds, has previously led to development of highly selective kinase inhibitors[45,46]. Closantel and Rafoxanide target the C-terminal domain allosteric sites of SPAK and OSR1, rather than their highly conserved ATP-binding pockets, leading to non-ATP-competitive kinase inhibition[44]. Iterative rounds of medicinal chemistry optimization led to identification of 5-chloro-N-(5-chloro-4-((4-chlorophenyl)(cyano)methyl)-2-methylphenyl)-2-hydroxybenzamide ("ZT-1a") as a selective SPAK inhibitor (Fig. 1 and Supplementary Table 1).

We compared the highest-potency salicylic amides selected from the library with Closantel, STOCK1S-14279, and STOCK1S-50699 as SPAK inhibitors in a cellular context. As NKCC1 and KCCs (KCC1–4) are phosphosubstrates of SPAK kinase[25,26], SPAK activity was monitored as NKCC1 Thr203/207/212 phosphorylation, which is required for cotransporter activation[36], and KCC site1/2 phosphorylation (KCC2 Thr906/Thr1007 or KCC3 Thr991/Thr1048), required for cotransporter inhibition[25,47] (Fig. 2, Supplementary Fig. 1). ZT-1a emerged as the most potent compound, inhibiting phosphorylation of NKCC1 p-Thr203/207/212 by $72 \pm 5.2\%$ at 1 μM ZT-1a and phosphorylation of KCC sites 1/2 by 65–77% at 3 μM (both $n = 4$, $p < 0.01$) in HEK-293 cells (Supplementary Table 1). SPAK phosphorylation at Ser373 was inhibited by $70 \pm 3.8\%$ inhibition at ~3–10 μM ZT-1a ($n = 3$, $p < 0.01$; Fig. 2). These results show ZT-1a to be a more potent modulator of SPAK-dependent CCC phosphorylation than the current SPAK kinase inhibitors Closantel, STOCK1S-50699, and STOCK1S-14279.

**Fig. 1 Hybrid design strategy yields the novel SPAK–CCC modulator ZT-1a.** Development of a new SPAK inhibitor (ZT-1a) through the combination of pharmacophores of the 2-(4-amino-2-chloro-5-methylphenyl)-2-(4-chlorophenyl)acetonitrile moiety from Closantel and the chloro-substituted 2-hydroxybenzoic acid from STOCK1S-14279 and Rafoxanide.

We next tested whether ZT-1a mediates kinase inhibition by competing for ATP binding to kinase in a manner similar to that of the nonspecific kinase inhibitor staurosporine. As shown in Supplementary Fig. 3a, staurosporine $IC_{50}$ values increased proportionally with increasing ATP concentrations, in contrast to the unchanged $IC_{50}$ values of ZT-1a (Supplementary Fig. 3b). These results suggest that ZT-1a inhibits SPAK kinase in a non-ATP-competitive manner. The kinase selectivity of ZT-1a was further assessed using standard radioisotopic enzymatic assays against a panel of 140 recombinant kinases (Dundee profiling, Supplementary Table 2)[48]. ZT-1a exhibits relatively high kinase selectivity, insofar as 98% of these 140 kinases were not inhibited by >70% by 1 μM ZT-1a as compared with DMSO control. Whereas 1 μM ZT-1a did inhibit GSK-3β activity by $60 ± 6\%$ compared in vitro with DMSO control, GSK-3β Ser9 phosphorylation was not inhibited by 3 μM ZT-1a either in HEK-293 cells (Supplementary Fig. 4a, b) or in ZT-1a-treated ischemic brain (Supplementary Fig. 4c, d).

**ZT-1a disrupts SPAK interaction with WNK but not with MO25α.** Crystallographic analysis of the human OSR1-conserved carboxyterminal (CCT) domain complexed to an RFXI motif-containing peptide derived from WNK[49] has shown that the highly conserved CCT residue Leu473 (mouse SPAK Leu502) forms critical hydrophobic contacts with the Phe residue of the RFXI motif. In vitro fluorescence polarization studies confirmed a 0.3-μM binding affinity of the RFXI motif-containing WNK peptide for purified wild-type SPAK protein, an interaction disrupted by STOCK1S-50699 with $IC_{50}$ of 2.51 μM (Supplementary Fig. 5). Neither Closantel nor ZT-1a disrupted binding between WNK4 and the SPAK CCT domain, suggesting that the ZT-1a binding site lies outside of the CCT domain (Supplementary Fig. 5).

In order to establish that ZT-1a binds to SPAK and OSR1 CCT domains, we employed a SPAK antibody pull-down assay from HEK-293 cell lysates in the presence of increasing ZT-1a concentrations, and compared the results with an 18-mer RFQV peptide from human WNK4[49] and an 18-mer AFQV negative control peptide not binding the primary pocket of SPAK and OSR1[49]. Co-immunoprecipitation of WNK1 with SPAK from HEK-293 cell lysates was abolished by ZT-1a more potently than by STOCK1S-50699 or by Closantel, in the absence or presence of competitor peptide (Supplementary Fig. 6). ZT-1a $IC_{50}$ values

were not significantly altered by the addition of MO25α, which activates SPAK/OSR1 ~100-fold and increases SPAK/OSR1-mediated in vitro phosphorylation of all CCCs[25,50] by ~8-fold (Supplementary Fig. 7).

**ZT-1a reduces SPAK-dependent CCC phosphorylation in cells.** To assess CCC phosphorylation changes in response to ZT-1a, HEK-293 cells were exposed for 30 min either to control isotonic medium or to hypotonic low [Cl−] medium (to activate SPAK/OSR1), and then treated with varying concentrations of ZT-1a for an additional 30 min. ZT-1a, in a dose-dependent manner (1–10 μM), markedly inhibited SPAK/OSR1 phosphorylation at Ser373/Ser325 (i.e., the activating site phosphorylated by WNK1) and NKCC1 phosphorylation at Thr203/Thr207/Thr212 (SPAK/OSR1 target sites whose phosphorylation is required for maximal transporter activity) in hypotonic low [Cl−] conditions as well as in control isotonic conditions (Fig. 2). These effects were paralleled by similar suppression of KCC3 Thr991/Thr1048 phosphorylation, consistent with WNK–SPAK/OSR1-mediated phosphorylation of these residues[25,26].

**ZT-1a inhibits NKCC1 but stimulates KCC3 activity.** $^{86}Rb^+$ flux assays have been utilized extensively as a reliable measurement of cation chloride cotransporter activity[51–54]. Therefore, the ability of ZT-1a to decrease inhibitory phosphorylation of KCC3 at Thr991/Thr1048 prompted us to assess ZT-1a's effect on KCC3 activity by measuring hypotonicity-stimulated $^{86}Rb^+$ uptake in isotonic or hypotonic low Cl− conditions (Fig. 3a). The furosemide (Furo)-treated cells, either transfected with empty vector or KCC3 wt cDNA, have significantly decreased K+ flux ($p < 0.01$; $n = 3$). Low KCC3 activity observed in wild-type (WT) HEK-293 cells was consistent with maximal KCC3 phosphorylation at Thr991/1048 in hypotonic low [Cl−] conditions[55]. In contrast, cells expressing KCC3 double mutant Thr991Ala/Thr1048Ala, mimicking activating dephosphorylation at these sites[56], exhibited a >13-fold increase in KCC3 activity compared with WT KCC3 ($p < 0.001$; $n = 3$). ZT-1a activated WT KCC3 activity >10.6-fold ($p < 0.001$; $n = 3$), but failed to increase activity of KCC3 double mutant Thr991Ala/Thr1048Ala ($p > 0.05$; $n = 3$; Fig. 3a), consistent with ZT-1a antagonism of SPAK-mediated phosphorylation at these sites.

As expected, given that NKCC1 is maximally phosphorylated following hypotonic low Cl− conditions, higher NKCC1 activity

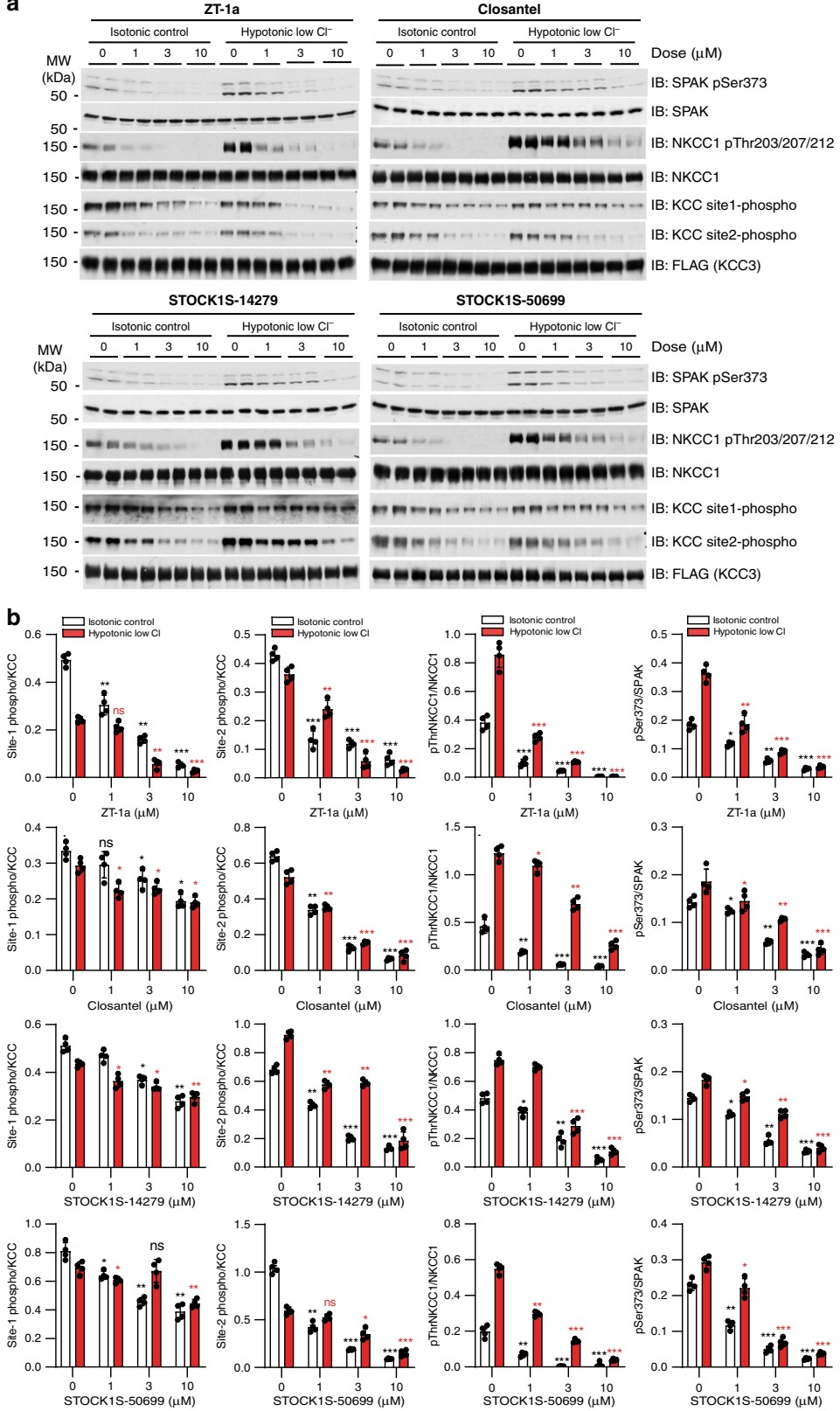

**Fig. 2 ZT-1a suppresses CCC phosphorylation. a** ZT-1a dose dependently inhibited KCC3 Thr991/Thr1048 phosphorylation in cells. HEK-293 cells were transfected with a DNA construct encoding wild-type N-terminally FLAG-tagged KCC3. Cells at 36 h post transfection were exposed for 30 min to either control isotonic or hypotonic low [Cl⁻] conditions, and then treated in the same conditions for an additional 30 min in the presence of inhibitors at the indicated concentrations. Cell lysates were subjected to SDS-PAGE and immunoblot with the indicated antibodies. **b** Immunoblot quantitation on ZT-1a, Closantel, STOCK1S-14279, and STOCK1S-50699. Band intensities were quantitated with ImageJ software. ***$p < 0.001$; **$p < 0.01$; *$p < 0.05$; ns, nonsignificant [by one-way ANOVA with post hoc testing ($n = 6$, error bars represent the mean ± SEM)].

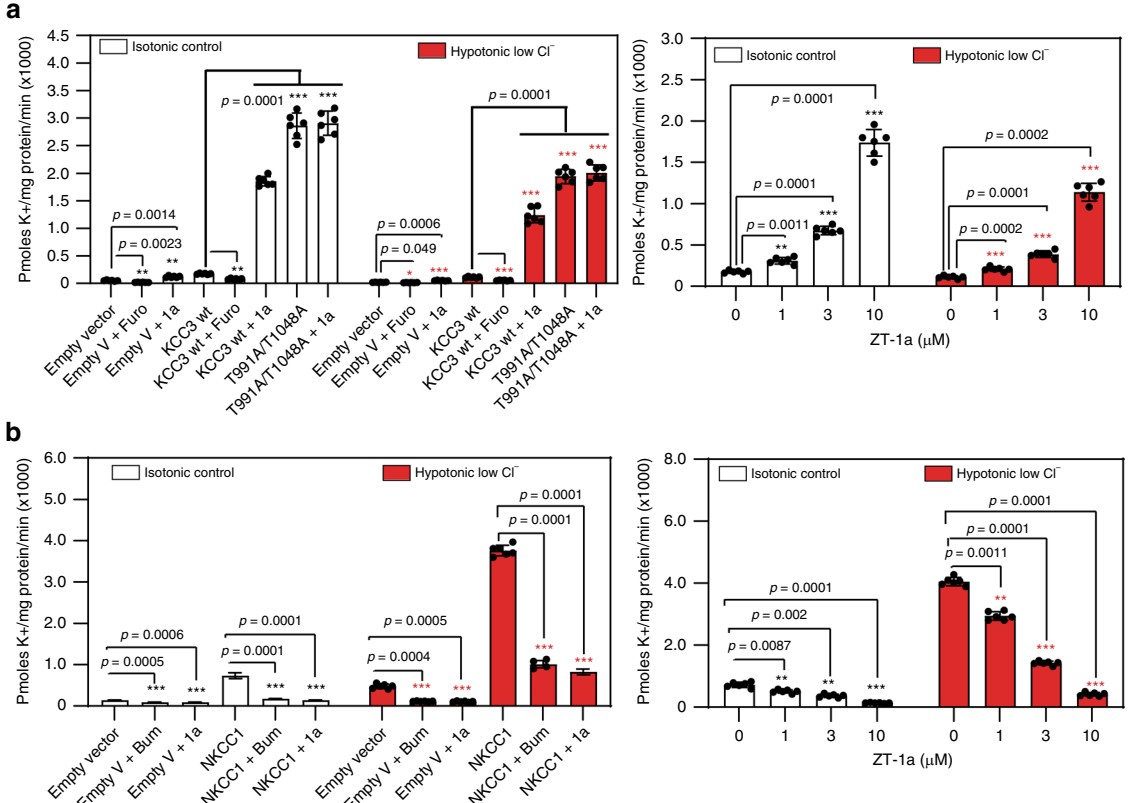

**Fig. 3 ZT-1a inhibition of SPAK kinase correlates with CCC activity. a** [86]Rb[+] uptake assays in the presence of ZT-1a measure transport activity of KCC3A. (**a**—left panel) HEK-293 cells were transfected with constructs encoding the indicated WT or mutant constructs of N-terminally FLAG-tagged KCC3. Thirty-six hours post transfection, cells were exposed for 30 min to either control isotonic conditions or hypotonic low [Cl[−]] conditions (to activate the SPAK/OSR1 pathway), and then treated in the same conditions for an additional 30 min with 10 μM ZT-1a or 1 mM furosemide (Furo) in the presence of 1 mM ouabain (Na[+]/K[+]-ATPase inhibitor) and 0.1 mM bumetanide (NKCC1 inhibitor). [86]Rb[+] uptake was allowed to proceed for 10 min and was then quantified by scintillation counting. [86]Rb[+] uptake counts per minute (CPM)s were normalized per mg protein and plotted for both isotonic and hypotonic conditions. (**a**—right panel) HEK-293 cells were transfected with constructs encoding wild-type N-terminally FLAG epitope-tagged KCC3A. Thirty-six hours post transfection, cells were exposed for 30 min to either control isotonic conditions or hypotonic low [Cl[−]] conditions (to activate the SPAK/OSR1 pathway), and then treated for an additional 30 min in the same conditions with the indicated ZT-1a concentrations in the presence of 1 mM ouabain and 0.1 mM bumetanide. Ten-minute [86]Rb[+] uptake values were quantified by scintillation counting, normalized per mg protein for each condition, and plotted for both isotonic and hypotonic conditions. ***$p < 0.001$; **$p < 0.01$; *$p < 0.05$; ns—nonsignificant [by one-way ANOVA with post hoc testing ($n = 6$, error bars represent the mean ± SEM)]. **b** [86]Rb[+] uptake assays in the presence of ZT-1a measure transport activity of NKCC1. (**b**—left panel) HEK-293 cells were transfected with WT NKCC1 cDNA construct. Thirty-six hours post transfection, cells were exposed for 30 min to either control isotonic conditions or hypotonic low [Cl[−]] conditions (to activate the SPAK/OSR1 pathway), and then treated in the same conditions for an additional 30 min with 10 μM ZT-1a or 0.1 mM bumetanide (Bum) in the presence of 1 mM ouabain (Na[+]/K[+]-ATPase inhibitor). [86]Rb[+] uptake was allowed to proceed for 10 min and was then quantified by scintillation counting. [86]Rb[+] uptake CPMs were normalized per mg protein and plotted for both isotonic and hypotonic conditions. (**b**—right panel) HEK-293 cells were transfected with WT NKCC1 cDNA construct. Thirty-six hours post transfection, cells were exposed for 30 min to either control isotonic conditions or hypotonic low [Cl[−]] conditions (to activate the SPAK/OSR1 pathway), and then treated for an additional 30 min in the same conditions with the indicated ZT-1a concentrations in the presence of 1 mM ouabain. Ten-minute [86]Rb[+] uptake values were quantified by scintillation counting, normalized per mg protein for each condition, and plotted for both isotonic and hypotonic conditions. ***$p < 0.001$; **$p < 0.01$; *$p < 0.05$; ns—nonsignificant [by one-way ANOVA with post hoc testing ($n = 6$, error bars represent the mean ± SEM)].

was observed. Cells treated with Bumetanide (Bum, an FDA-approved potent loop diuretic (LD) that acts by antagonizing NKCC1/NKCC2), whether transfected with empty vector or with NKCC1 wt cDNA, have significantly decreased K[+] flux in isotonic or in hypotonic low Cl[−] conditions ($p < 0.001$; $n = 3$; Fig. 3b). In contrast, NKCC1-mediated K[+] influx in isotonic or in hypotonic low Cl[−] conditions was significantly inhibited by ZT-1a at concentration > 1 μM ($p < 0.01$; $n = 3$; Fig. 3b). These results demonstrated that ZT-1a stimulates KCC3 activity but inhibits NKCC1 activity.

**ZT-1a inhibits SPAK-dependent CCC phosphorylation in vivo.** To define ZT-1a efficacy in naive mice in vivo, we examined

phosphorylation of SPAK, NKCC1, and NCC in kidney, and phosphorylation of SPAK, NKCC1, and KCC3 in brain, after intraperitoneal (i.p.) administration of either ZT-1a (10, 30, 50, and 100 mg/kg) or Closantel (50 mg/kg as a reference). NCC phosphorylation (pThr46/50/55/60) and NKCC1 phosphorylation (pThr203/207/212) were reduced ~83–90 ± 7.9% in naive WT kidneys 30 min after ZT-1a administration (50 mg/kg; $p < 0.01$; $n = 3$; Supplementary Fig. 8b). In contrast, at 60 min after administration of ZT-1a or Closantel, neither drug had reduced KCC3 phosphorylation (pThr1048) or NKCC1 phosphorylation (pThr203/207/212) in naive brains (Supplementary Fig. 8a). SPAK Leu502 is a residue required for high-affinity recognition of the RFXV/I motifs in SPAK upstream activator WNK isoforms, and SPAK[502A/502A] mice served as controls for true

corresponding residues in SPAK[49,57]. KCC3 phosphorylation at pThr1048 was reduced $34 \pm 4.6\%$ in brains from SPAK[502A/502A] mice ($p < 0.01$; $n = 3$; Supplementary Fig. 8a), a model of Gitelman syndrome[57]. Thus, systemically administered ZT-1a has low efficacy in naive brains, suggesting inefficient ZT-1a transport across the BBB, likely in part reflecting unfavorable ZT-1a pharmacokinetics in naive mice ($T_{1/2} = 1.8$ h, AUC = 2340 h*ng/mL and %F = 2.2%; Supplementary Table 3).

**ZT-1a reduces CSF secretion in hemorrhagic hydrocephalus.** Choroid plexus epithelium (CPe) is the most actively secreting epithelium in the body[38,58], producing up to 500 cc/day of CSF[59]. Among all tissues, including kidney[57], SPAK kinase is most highly expressed in CPe[3]. Several CCCs are also expressed in choroid plexus[60–62], including NKCC1, recently shown to be essential for CSF secretion[1,3]. IVH-triggered TLR4 signaling stimulates CSF secretion >3.5-fold and causes hydrocephalus by increasing functional expression of pSPAK and pNKCC1 in CPe[3]. We speculated that intracerebroventricular (ICV) delivery of ZT-1a into the cerebrospinal fluid might bypass the blood–brain barrier and allow ZT-1a to exert its effects on SPAK and CCCs in CPe.

To assess KCC1–4 protein expression in rat choroid plexus, we first verified isoform specificity of our antibodies. Flag-tagged KCC1–4 were individually expressed in HEK-293 cells. The purified fraction of KCC immunoprecipitated by KCC site-2 phospho-antibody was subjected to SDS-PAGE and immunoblot with each antibody. As evident from Fig. 4a, each antibody recognized only its respective target, without isoform cross-reactivity. Although immunospecific, none of these antibodies were able to detect endogenous KCCs from rat CPe lysates through direct immunoblot. However, immunoblot of immunoprecipitated fractions demonstrated robust immunoreaction with antibodies targeting KCC1, KCC3, and KCC4, whereas CPe KCC2 abundance remained below the detection limit, consistent with its documented neuron-specific expression pattern[63].

We next tested the effects of ICV delivery of ZT-1a on pSPAK, pNKCC1, pKCC1, pKCC3, and pKCC4 in rat CPe in the setting of experimental hemorrhagic hydrocephalus, as described[3]. ZT-1a reduced IVH-induced expression of pSPAK by $55 \pm 3.6\%$; $p < 0.01$; $n = 3$ and pNKCC1 by $69 \pm 4.3\%$; $p < 0.001$; $n = 3$ (Fig. 4b). ZT-1a reduced post-IVH KCC1 phosphorylation at Thr927/Thr983 by 12- and 3.4-fold, respectively ($p < 0.001$; $n = 3$), KCC3 phosphorylation at Thr991/Thr1048 by 6- and 2.1-fold, respectively ($p < 0.001$; $n = 3$), and KCC4 phosphorylation at Thr926/Thr980 (corresponding to KCC3 Thr991/Thr1048) by ~2-fold ($p < 0.001$; $n = 3$) in CPe (Fig. 4b). Consistent with these results, ZT-1a treatment for 48 h at 10 μM decreased post-IVH CSF hypersecretion by ~2.3-fold ($p < 0.01$), in contrast to the lack of effect of DMSO vehicle ($p > 0.05$) (Fig. 4c). These data suggest that ICV administration of ZT-1a can effectively modulate pathological CSF secretion by decreasing SPAK–NKCC1/KCC phosphorylation.

**ZT-1a modulates primary cortical neuron volume.** SPAK–NKCC1 signaling plays an important role in regulating cell volume in response to hypertonic osmotic stress. The effects of the SPAK inhibitor ZT-1a and the NKCC1 inhibitor bumetanide (BMT) on cell volume regulation were assessed in primary cortical neurons exposed to hyperosmotic stress (370 mOsm/kg $H_2O$) (Fig. 5a). As shown in Fig. 5b, c, ~30% neurons (16/49 cells) in control conditions showed regulatory volume increase (RVI) at a rate of $0.0256 \pm 0.006$ relative volume change/min in response to hypertonicity-mediated osmotic shrinkage ($p < 0.0001$). However, in the presence of either BMT or ZT-1a, neurons displayed

gradual shrinkage with no RVI response, suggesting roles of SPAK–NKCC1 complex in counteracting cell shrinkage. Upon returning to isotonic conditions after hyperosmotic stress, the drug-treated neurons recovered their volume rapidly (Fig. 5b). The similar effects mediated by BMT and ZT-1a treatments suggest that they target the same signaling cascade. The absence of RVI was also observed in human fibroblast cells treated with BMT or ZT-1a (Supplementary Fig. 9).

**ZT-1a reduces stroke-associated cerebral edema and infarct.** Ischemic stroke is associated with significant upregulation of SPAK and NKCC1 phosphorylation in peri-infarct cortex, striatum, and corpus callosum[39]. Genetic inhibition of either SPAK or NKCC1 decreases ischemic cerebral edema and improves neurological outcomes[42,64]. We evaluated ZT-1a efficacy in attenuating cerebral infarct and associated cerebral edema in a mouse model of ischemic stroke[39,42]. First, we examined intrinsic immunofluorescence labeling of serum albumin at 24 h after tMCAO, as an indication of BBB integrity. As shown in Fig. 6a–c, serum-albumin extravasation into brain parenchyma was ~5-fold higher in mice with ischemic stroke than in sham-operated mice ($p < 0.05$, $n = 5$–6), suggesting that tMCAO/reperfusion induced disruption of BBB integrity in ischemic brains. We then assessed whether ZT-1a penetrates better in ischemic brains due to the leaky BBB. Two hours after ZT-1a administration (5 mg/kg i.p.), plasma ZT-1a levels in ischemic and non-ischemic sham mice were indistinguishable ($p = 0.29$; $n = 10$–11, Fig. 6e), whereas ZT-1a concentration in both CL and IL hemispheres of ischemic brains was ~1.8-fold higher than that in sham brains ($p = 0.006$; $n = 10$–11, Fig. 6f). Importantly, ZT-1a administration 3 h post reperfusion decreased infarct volume by ~44% (from $78.1 \pm 5.8$ mm$^3$ in the vehicle-control group to $43.6 \pm 6.5$ mm$^3$, $p < 0.01$, $n = 12$, Fig. 7b). Moreover, ZT-1a treatment at either 2.5 or 5 mg/kg reduced cerebral hemispheric swelling by ~36–54% as compared with vehicle control ($p < 0.01$, Fig. 7b). Regional cerebral blood flow (rCBF) within the 24-h post-stroke period was unaffected by ZT-1a (Supplementary Fig. 10a–c). These results confirm that ZT-1a reduces infarct size and ischemic cerebral edema.

**ZT-1a improves neurological function after ischemic stroke.** We evaluated the impact of ZT-1a on progression of sensorimotor function deficits in a mouse model of ischemic stroke. The vehicle-control mice developed persistent, severe neurological deficits at days 0–7 after stroke, as reflected in elevated neurological scores of 2.5–2.9 (Fig. 7c). ZT-1a-treated mice exhibited a progressive decrease in neurological deficit scores between day 1 ($2.0 \pm 0.2$) and day 7 ($1.4 \pm 0.2$, $p < 0.01$, $n = 6$). In the corner test[42] evaluating post-ischemic sensory and motor deficits, vehicle-control mice exhibited behavioral asymmetries 1 day after stroke, deficits that persisted for 7 days (Fig. 7c). In contrast, ZT-1a-treated mice showed reduced unidirectional turn preference and absence of asymmetries by day 7 post stroke. In the adhesive contact and removal tests that evaluate fine sensorimotor function deficits[65], ZT-1a-treated mice displayed significantly improved motor function ($p < 0.05$, $n = 6$) as compared with vehicle-control mice. These data show that post-stroke treatment with ZT-1a significantly improved mouse neurological functional recovery.

**ZT-1a protects gray and white matter after ischemic stroke.** We conducted ex vivo MRI studies of brains from the same cohort of vehicle-control and ZT-1a-treated mice after completion of their neurological functional assessment at 7 days post stroke. T2-weighted MRI analysis further confirmed that ZT-1a treatment

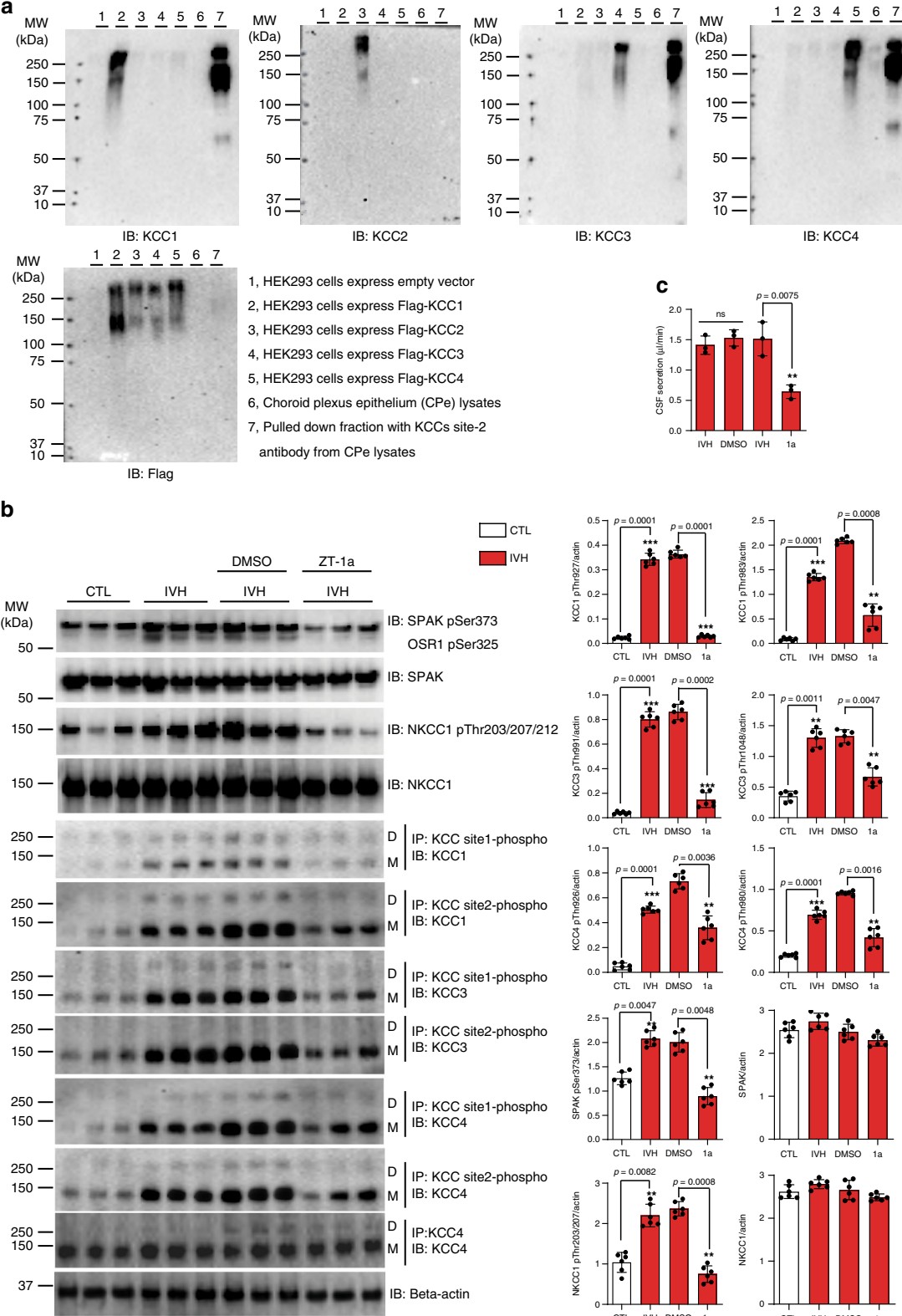

reduced stroke-induced lesion volume by ~40% and longer-term brain atrophy (% hemisphere shrinkage) by ~41% ($p < 0.05$, $n = 6$, Fig. 7d). To assess the effect of ZT-1a on stroke-induced white matter injury, we analyzed fractional anisotropy (FA) and directionally encoded color (DEC) maps of the corpus callosum (CC) and external capsule (EC) in the vehicle-control and ZT-1a-treated brains. DEC and FA maps revealed intact CC and EC

tracks in the CL hemisphere (arrows) and injured EC in the IL hemisphere (arrowheads) (Fig. 7d). FA values were significantly reduced in the ipsilateral EC tract of the vehicle-control brains, indicating loss of white matter integrity after stroke. In contrast, EC tract FA values were undiminished in the IL hemisphere of ZT-1a-treated mice ($p > 0.05$, $n = 6$), and the FA values were higher than the IL hemisphere in vehicle-control brains ($p < 0.05$),

**Fig. 4 ZT-1a decreases SPAK–CCC phosphorylation in hemorrhagic hydrocephalus. a** Representative immunoblots verify antibody specificity. KCC1–4 were expressed in HEK-293 cells. KCC site-2 phospho-antibody pull-down fractions were subjected to SDS-PAGE followed by immunoblot with antibodies targeting specific transporters ($n = 3$). **b** Effect of ZT-1a (ICV, 10 mmol) on IVH-induced phosphorylation of SPAK/OSR1, NKCC1, and KCC4 in CPE, as measured in control rats (CTL) and in rats 48 h post IVH treated with ZT-1a or vehicle ($n = 3$). CPE lysates were harvested and subjected to immunoprecipitation (IP) and immunoblot (IB) with the indicated antibodies. D, dimeric KCCs; M, monomeric KCCs. Quantitation of actin-normalized choroid plexus ion transporter phosphorylation is presented in control rats (CTL) and rats 48 h after experimental IVH in the presence or absence of ZT-1a or vehicle ($n = 3$). ***$p < 0.001$; **$p < 0.01$; *$p < 0.05$ versus control rats [by one-way ANOVA with post hoc testing (error bars represent the mean ± SEM)]. **c** Effect of SPAK inhibition by 10 μM ZT-1a on IVH-induced CSF hypersecretion by CPE ($n = 3$). The rate of CSF production (μL/min) is presented as means ± SEM. ***$p < 0.001$; **$p < 0.01$ versus baseline IVH [one-way ANOVA with post hoc testing (error bars represent the mean ± SEM)].

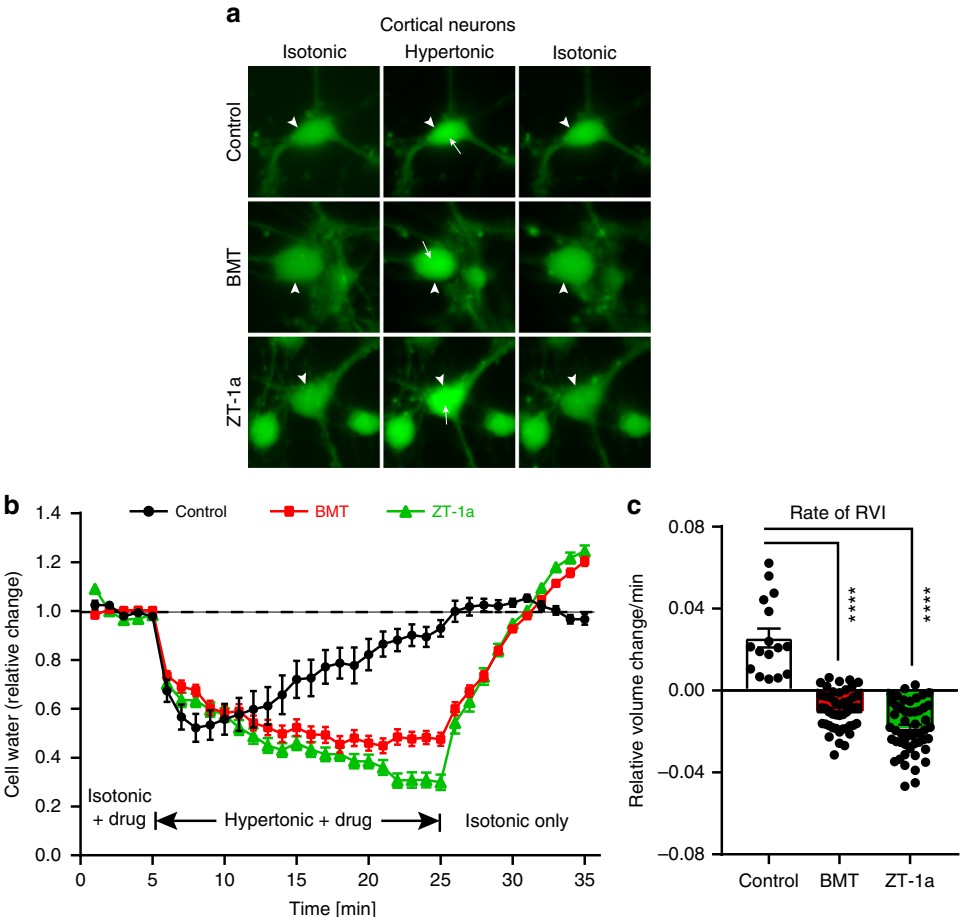

**Fig. 5 Effects of ZT-1a on cell volume changes in mouse primary cortical neurons. a** Representative images of calcein-AM-loaded mouse primary cortical neurons. Arrowheads mark cells of interest. Arrows illustrate regions of increased fluorescence intensity in response to hyperosmotic stress. Cells were exposed to isotonic HEPES-MEM (310 mOsm, 5 min, ± drug), hypertonic HEPES-MEM (370 mOsm, 20 min, ± drug), and isotonic HEPES-MEM (10 min, no drug) in the absence or presence of SPAK inhibitor ZT-1a (3 μM) or the potent NKCC1 inhibitor bumetanide (BMT, 10 μM). **b** Relative cell water content changes as a function of time during indicated bath conditions. **c** Summary data of the rate of regulatory volume increase (RVI). The rate was calculated by determining the slope between minutes 8 and 25. Data are means ± SEM, $n = 49$ cells for control, 46 cells for BMT, and 49 cells for ZT-1a collected from three independent experiments. ****$p < 0.0001$ versus control, one-way ANOVA.

reflecting ZT-1a-mediated preservation of EC white matter microstructure. These results further suggest that post-stroke treatment with ZT-1a protects both gray and white matter tissues in ischemic brains.

**ZT-1a inhibits stroke-induced SPAK–CCC phosphorylation.** Ischemia elicits neuronal and glial cell swelling[66]. In contrast to the neuron-specific KCC2, NKCC1 and KCC3 are expressed in both neurons and glia[67,68], and play roles in cell volume regulation[69]. We therefore examined ZT-1a effects on phosphorylation of SPAK/OSR1, NKCC1, and KCC3 in ischemic mouse brains. Ischemic stroke increased phosphorylation of pSPAK (pSer373)/pOSR1 (pSer325) by ~1.5-fold ($p < 0.05$, $n = 6$), pNKCC1 (pThr203/207/212) by ~1.6-fold ($p < 0.05$, $n = 6$), and pKCC3 pThr991 ($p < 0.05$, $n = 6$) and pThr1048 by ~1.3-fold (both $p < 0.05$) in membrane protein fractions from the ipsilateral (IL) cortical hemisphere at 24 h of reperfusion in vehicle-control-treated mice, without significant change in the corresponding total protein levels (Fig. 8a, b). Post-stroke administration of ZT-1a in mice prevented ischemia-induced increases of pSPAK/pOSR1, pNKCC1, and pKCC3 ($p < 0.05$, $n = 6$) without alteration in the corresponding total protein expression

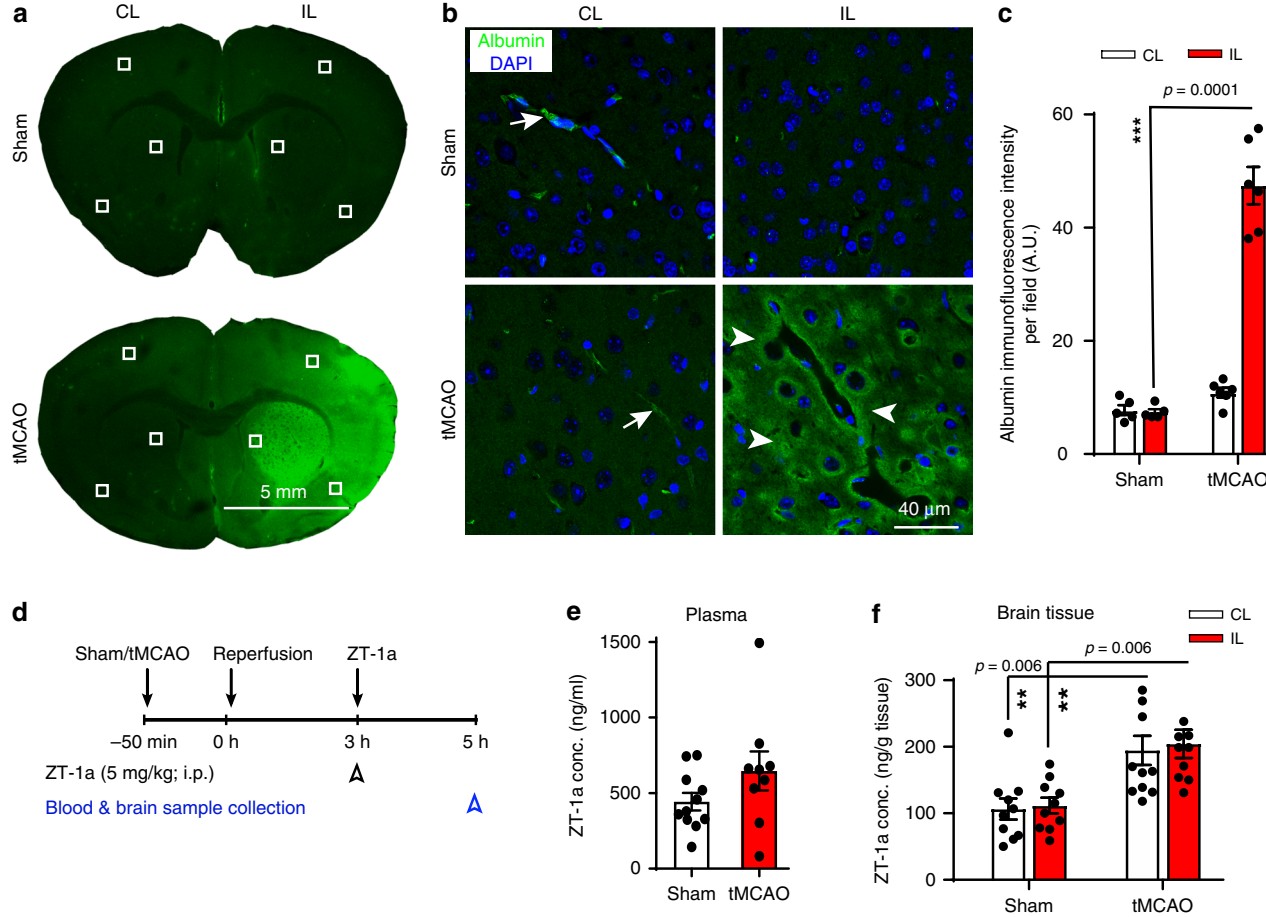

**Fig. 6 Ischemic stroke facilitates ZT-1a entrance into the brain. a** Anti-serum-albumin-labeled brain sections at 24 h after sham or tMCAO surgery illustrate image collection in the perilesion and contralateral (CL) control areas (white boxes). Note that tMCAO-subjected mice exhibited increased serum-albumin immunofluorescence intensity in the ipsilateral (IL) hemisphere compared with that of sham mice. **b** Higher-magnification (×40) images of serum-albumin immunofluorescence. The arrow shows albumin in vessel lumen. The arrowhead shows albumin that has diffused into ischemic brain parenchyma. **c** Quantitative analysis of albumin immunofluorescence intensity as shown in **b**. Data are mean ± SEM, $n = 5$ per group (male); ***$p < 0.0001$, one-way ANOVA. **d** Experimental design for bioavailability assay of ZT-1a in mouse plasma and brain. ZT-1a was administered at 3 h post reperfusion or post sham surgery. Blood and brain samples were collected at 2 h post injection. **e** Plasma ZT-1a concentrations were indistinguishable in sham-operated and tMCAO-subjected male mice ($p = 0.29$; $n = 11$ (sham) and 9 (tMCAO)). **f** Increased ZT-1a concentration was detected in contralateral (CL) and ipsilateral (IL) brain tissues of ischemic versus sham-operated male mice. Data are mean ± SEM, $n = 10$ (sham) and 11 (tMCAO). **$p < 0.01$, one-way ANOVA.

(Fig. 8a, b; Supplementary Fig. 11). These results indicate that ZT-1a inhibits the SPAK-dependent upregulation of NKCC1 and KCC3 phosphorylation in ischemic brains.

**Closantel reduces ischemic brain injury.** We compared the neuroprotective efficacy of Closantel with that of the pan-WNK-kinase inhibitor WNK463. Administration of Closantel (1.0 or 2.5 mg/kg) at 3 and at 8 h post reperfusion reduced infarct volume and hemispheric swelling in a dose-dependent manner ($p < 0.01$, $n = 5–14$, Fig. 9a–c), whereas Closantel at 0.1 mg/kg was ineffective. As was true for ZT-1a, Closantel did not alter rCBF (Supplementary Fig. 9a–f). In contrast, WNK463 treatment (2.5 mg/kg) reduced neither infarct volume nor hemispheric swelling at 24 h post reperfusion (Fig. 9a–c). Moreover, toxicity of Closantel and ZT-1a at 25 mg/kg in naive and ischemic stroke mice was reflected in Kaplan–Meier survival curves (Supplementary Fig. 12). Ischemic stroke mice treated with ZT-1a survived longer than Closantel-treated mice at this high dose ($p = 0.008$, $n = 6$, log-rank test). These data demonstrate that ZT-1a and Closantel are effective for in vivo inhibition of the WNK–SPAK–CCC pathway in the mouse model of ischemic brain injury.

## Discussion

We have applied a "scaffold-hybrid" strategy to discover and characterize a potent and selective inhibitor of SPAK kinase, ZT-1a [5-chloro-$N$-(5-chloro-4-((4-chlorophenyl)(cyano)methyl)-2-methylphenyl)-2-hydroxybenzamide]. About 3 μM ZT-1a inhibited stimulatory phosphorylation of SPAK Ser373 and NKCC1 Thr203/Thr207/Thr212, and the inhibitory phosphorylation of the KCCs (e.g., KCC3 Thr991/Thr1048). As shown in cellular $^{86}Rb^+$ flux assays, ZT-1a inhibited NKCC1 but potently stimulated KCC activity >10-fold. In a model of post-hemorrhagic hydrocephalus, intracerebroventricular delivery of ZT-1a decreased choroid plexus SPAK–CCC phosphorylation and antagonized inflammation-induced CSF hypersecretion. In a model of ischemic stroke, systemic ZT-1a decreased neuronal and glial CCC phosphorylation, reduced infarct size and cerebral edema, and improved neurological outcomes. These results suggest that ZT-1a is an effective SPAK–CCC modulator, with therapeutic potential for treatment of brain disorders associated with impaired cell volume homeostasis.

The traditional strategy of targeting the ATP-binding site of SPAK/OSR1 increases risks of off-target kinase inhibition. The

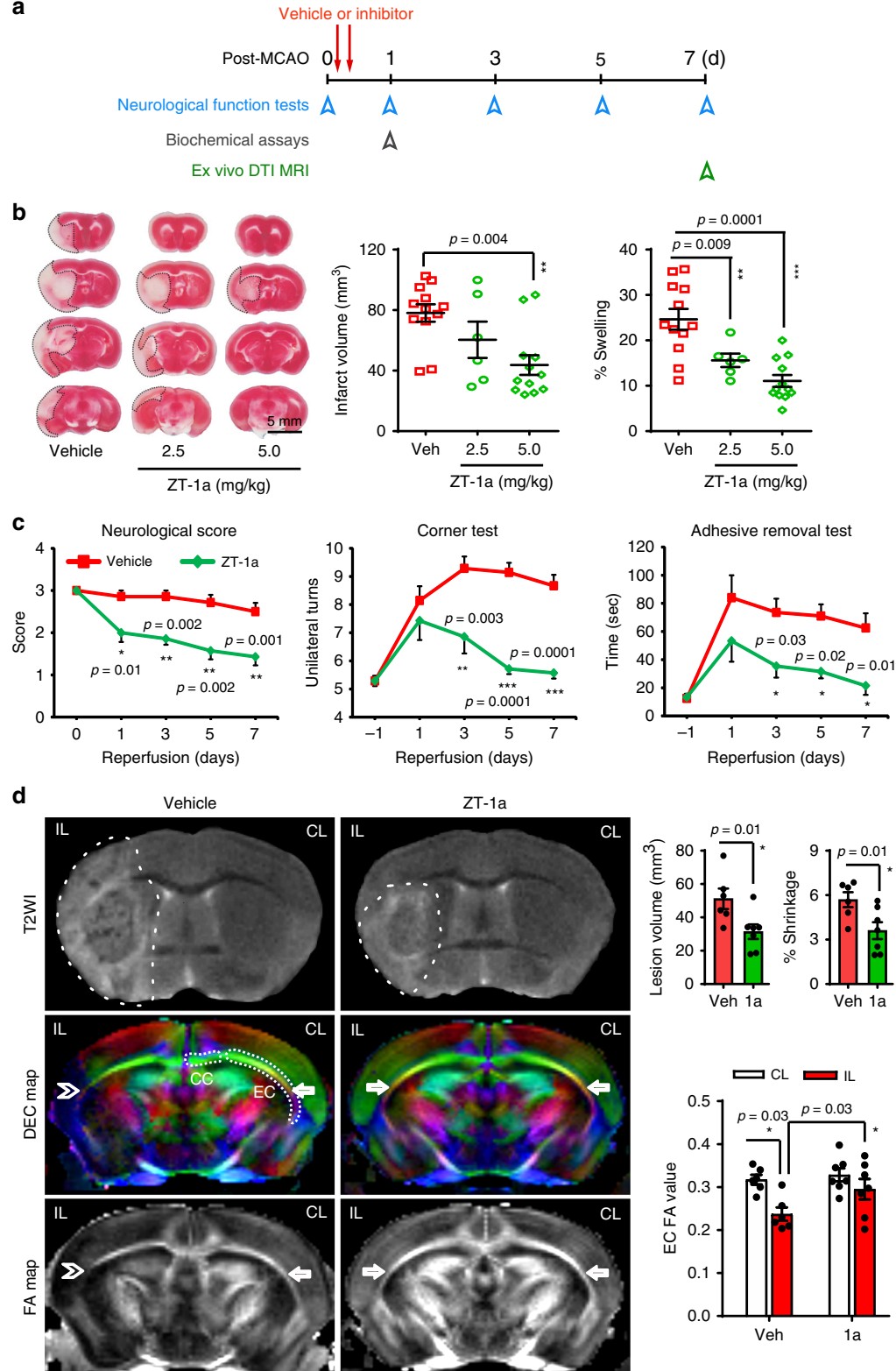

introduction of STOCK1S-50699 and STOCK2S-26016[70] has highlighted SPAK inhibitors that target the CCT domain rather than the kinase domain. However, only STOCK1S-50699, but not STOCK2S-26016, suppressed in vitro phosphorylation of SPAK/OSR1 and NKCC1 induced by hypotonic low [Cl⁻][25]. Moreover, in vivo pharmacokinetics of STOCK1S-50699 are unfavorable (data not shown). The livestock antiparasitic drug, Closantel[71], emerged as the first drug candidate for in vivo pharmacological inhibition of SPAK[43,44], but is contraindicated in humans due to retinal toxicity[72].

The ATP independence of SPAK inhibition by Closantel and STOCK1S-14279 has suggested the possibility of developing inhibitors of WNK signaling by binding to constitutively active or WNK-insensitive (T233E) SPAK[43]. With a "scaffold-hybrid"

**Fig. 7 ZT-1a decreases brain injury after ischemic stroke. a** Experimental design of ischemic stroke study. Vehicle or ZT-1a is administered at 3 and 8 h post tMCAO. **b** Representative images and quantitation of infarct volume and hemispheric swelling in TTC-stained coronal sections of mouse brains 24 h post tMCAO. Vehicle (DMSO, 2 ml/kg) or ZT-1a (2.5 or 5.0 mg/kg) were administered via intraperitoneal injection (i.p.) with initial half-dose at 3 h and second half-dose at 8 h post reperfusion. Data are mean ± SEM, male mice, $n = 12$ (vehicle), 6 (ZT-1a 2.5), and 12 (ZT-1a 5.0). ***$p < 0.001$; **$p < 0.01$ versus vehicle-control mice, one-way ANOVA. **c** Neurological deficit scores, corner tests, and adhesive tape removal tests of mice 1 day before tMCAO (−1) and at days 0, 1, 3, 5, and 7 post tMCAO. Vehicle (DMSO, 2 ml/kg) or ZT-1a (5.0 mg/kg) were administered as described in **b**. Data are means ± SEM, $n = 7$ for each group (male 4, female 3). ***$p < 0.001$; **$p < 0.01$; *$p < 0.05$ versus the respective vehicle control. **d** Representative images of $T_2$WI, directionally encoded color (DEC), and fractional anisotropy (FA) maps of ex vivo brains from the same cohort of mice at 7 days post tMCAO, as described in **c**. Arrow marks EC (external capsule); double arrowhead marks damaged EC; CC: corpus callosum. Bar graphs display quantitation of brain lesion volume, brain atrophy (% shrinkage), and mean FA values. Data are means ± SEM, $n = 6$ for vehicle (male 3, female 3) and 7 for ZT-1a (male 4, female 3). *$p < 0.05$, one-way ANOVA.

strategy[45,46], we designed and synthesized a new, focused chemical library derived from these two ATP-insensitive inhibitors. ZT-1a was among the best SPAK inhibitors identified from an ~300-compound chemical library, and was characterized by fluorescence polarization, in vitro kinase, and cell-based assays. ZT-1a was a more potent CCC inhibitor than its predecessor SPAK inhibitors, Closantel[43], Rafoxanide[44], STOCK1S-14279[43], and STOCK1S-50699[70]. About 3 μM ZT-1a substantially inhibited phosphorylation of SPAK Ser373, NKCC1 Thr203/207/212, and KCC3 Thr991/Thr1048 (KCC4 Thr926/Thr980) in cells. ZT-1a also effectively inhibited NKCC1 and stimulated KCC3 in standard cellular assays of CCC-mediated $^{86}$Rb$^+$ flux.

Post-stroke administration of Closantel in mice reduced ischemic cerebral infarction size and brain swelling in a dose-dependent manner. However, we detected Closantel toxicity in mice at higher doses (~25 mg/kg), consistent with reported adverse effects in humans, including weakness, visual impairment, and blindness[72,73]. The in vitro nanomolar-potency pan-WNK-kinase inhibitor WNK463 was developed as an anti-hypertensive drug[74]. Oral administration of 1–10 mg/kg WNK463 reduced blood pressure and regulated body fluid and electrolyte homeostasis in normotensive and hypertensive rodent models[74]. However, 2.5 mg/kg of WNK463 in mice was not neuroprotective, but elicited ataxia and breathing difficulties, as previously reported at 1–10 mg/kg doses[74]. Although the causes of these adverse effects remain unclear, they also discouraged efficacy testing of WNK463 at higher doses.

Post-stroke administration of ZT-1a attenuated stroke-associated infarction and cerebral edema and rapidly improved neurological function in post-ischemic mice, the results corroborated by brain MRI. T2-weighted MRI of ex vivo brains from ZT-1a-treated mice showed smaller lesion volumes and reduced brain atrophy at 7 days post stroke. DTI data analysis revealed that ZT-1a significantly prevented subacute brain white matter injury after ischemic stroke. ZT-1a thus provides neuroprotection, at least in part by directly inhibiting SPAK kinase activity and SPAK-mediated phosphoactivation of NKCC1 and phosphoinactivation of KCC3 in ischemic brains. ZT-1a reduced ischemia-induced elevations of pSPAK and pNKCC1 by 55–65%, and elevations of pKCC3 by 30% compared with vehicle-control mice. The ZT-1a-mediated concurrent inhibition of phosphorylation of SPAK and of NKCC1/KCC3 supports our interpretation that ZT-1a exhibits allosteric inhibition of SPAK kinase activity likely through binding to the secondary pocket of the CCT domain, thus preventing SPAK binding and activation by upstream WNK kinases. A similar dual mechanism of WNK–SPAK inhibitor STOCK1S-50669 has been reported in cultured cells[44,70].

Plasma half-life of ZT-1a is only ~1.8 h in normal naive mice, and ZT-1a penetration across the healthy BBB is minimal. Ischemic stroke injury disrupts BBB tight junctions as early as 30 min after resumption of perfusion[75,76], facilitating brain access

for many small-molecule drugs[77]. We have shown that ischemic stroke caused robust extravasation of serum albumin into brain parenchyma, in parallel with detection of elevated ZT-1a levels in ischemic brain. These findings suggest that ZT-1a more effectively entered ischemic brain parenchyma across a leaky BBB. However, we cannot rule out possible contributions from altered BBB drug transporter pathways and/or reduced ZT-1a extrusion from CNS in the setting of ischemia. Solute carrier (SLC) transporters, such as organic anion-transporting polypeptides (Oatps), may improve drug delivery into the brain, whereas the ATP-binding cassette (ABC) transporter superfamily (P-glycoprotein) exports drugs across BBB endothelial cells into the blood[78,79]. As ischemia increases Oatp1a4 protein expression and function in brain microvessels[80], this and other SLC proteins of similar expression pattern may contribute to increased ZT-1a transport across BBB into ischemic brains. Whether ZT-1a is a substrate for Oatp1a4 and ABC superfamily remains to be studied. Ischemic stroke-induced disruption of the BBB integrity is also evident in human brains[81,82], suggesting that small-molecule neuroprotective drugs, such as ZT-1a, could potentially traverse the leaky BBB into ischemic human brain. We administered the initial dose of ZT-1a 3-h post stroke, a time comparable to the 2.5–4.0-h post-stroke treatment windows found effective for the candidate neuroprotective agents, glibenclamide, human albumin, and minocycline[83–85]. Although ZT-1a and the antiparasitic drug Closantel exhibit similar effects in reducing ischemic brain damage, Closantel shows toxicity in mouse and human[72,73]. In this study, we did not test possible visual system toxicity of ZT-1a. Additional pharmacokinetics and toxicity studies of ZT-1a, especially on the retina, warrant further investigations for future clinical translation.

The CPe secretes ~500 ml/day CSF, a higher per-cell fluid volume than for any other epithelium[38]. NKCC1 expressed in the apical CPe contributes approximately half of CSF production, though its mechanism remains unclear, in view of its unique apical localization in the CPe compared with its basolateral position in all other epithelia[1]. Indeed, some have proposed inward NKCC1 flux to be required for apical K$^+$ recycling and continued CSF production[86,87], while others have provided evidence for net ion efflux with obligatory water transport directly contributing to CSF production[1,3]. Nonetheless, in a rat model of hemorrhagic hydrocephalus, intraventricular hemorrhage causes a Toll-like receptor 4 (TLR4)- and NF-κB-dependent inflammatory response in CPe associated with ~3-fold increase in bumetanide-sensitive CSF secretion[3]. IVH-induced hypersecretion of CSF is mediated by TLR4-dependent activation of SPAK, which binds, phosphorylates, and stimulates NKCC1 at the CPe apical membrane[3]. Genetic depletion of TLR4 or SPAK normalizes hyperactive CSF secretion rates and reduces post-hemorrhagic hydrocephalus symptoms in parallel with reduction of NKCC1 phosphorylation. We have shown here that ICV administration of ZT-1a restores CSF secretion rates to basal

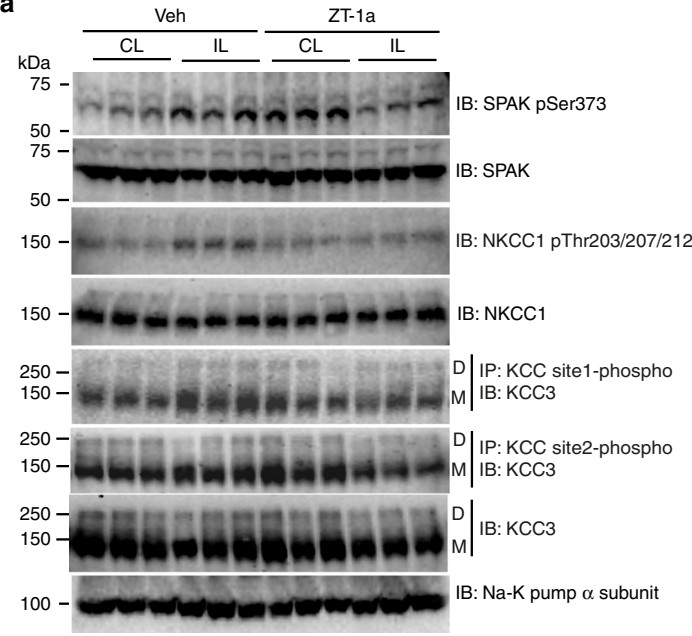

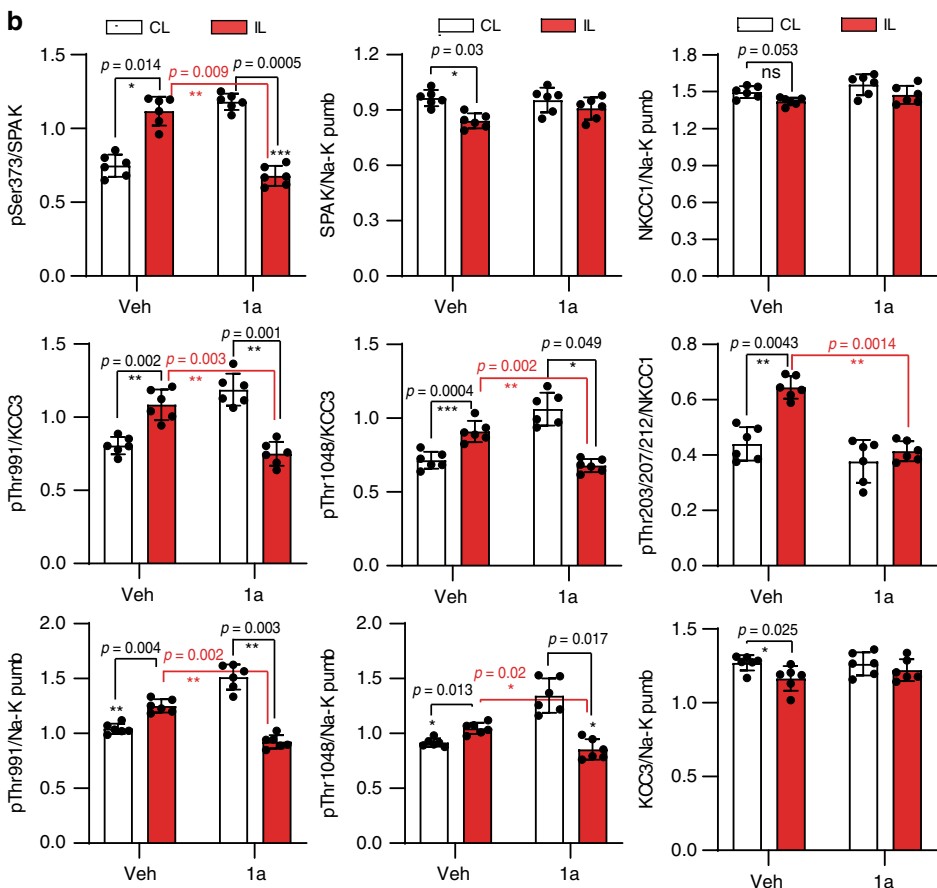

**Fig. 8 ZT-1a decreases ischemia-induced SPAK–CCC phosphorylation. a** Representative immunoblots (IB) of phospho-SPAK/OSR1 (pSPAK/pOSR1), phospho-KCC3 (pKCC3), and phospho-NKCC1 (pNKCC1) in mouse brains studied 24 h post reperfusion after ischemic stroke. Membrane protein fractions were prepared from contralateral (CL) and ipsilateral (IL) cerebral hemispheres. Vehicle (Veh, DMSO) or ZT-1a (5 mg/kg) were administered as described in Fig. 7a. $Na^+–K^+$ ATPase α-subunit served as loading control for membrane protein fraction. **b** Densitometry analyses of immunoblots (similar to those in panel **a**) of pSPAK/pOSR1, pKCC3, pNKCC1, tSPAK/tOSR1, and tNKCC1 in mouse brains studied 24 h of reperfusion after tMCAO. Data are means ± SEM, $n = 6$ per group (male 3, female 3). ***$p < 0.001$; **$p < 0.01$; *$p < 0.05$ versus control, one-way ANOVA.

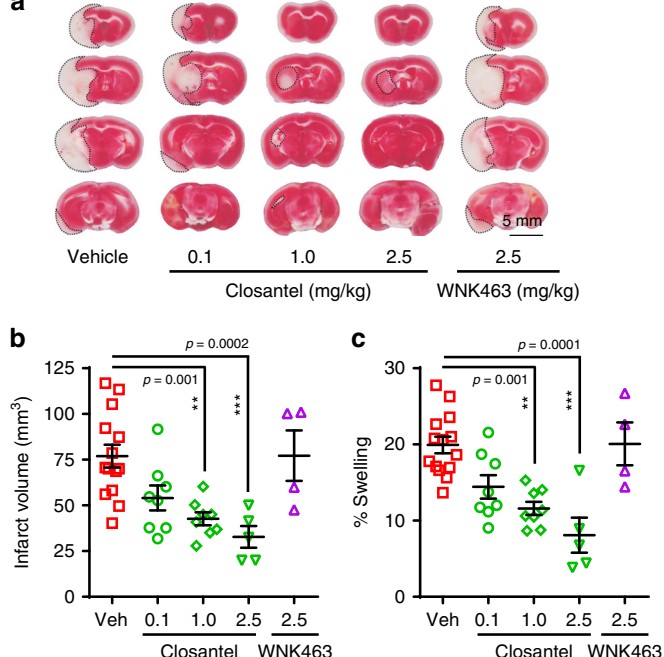

**Fig. 9 The SPAK inhibitor Closantel exhibits dose-dependent neuroprotective effects. a** Representative images of TTC-stained coronal sections of mouse brain 24 h post tMCAO. Vehicle (Veh) control (DMSO, 2 ml/kg), Closantel (0.1, 1.0, or 2.5 mg/kg), or WNK463 (2.5 mg/kg body weight) were administered i.p. in divided doses at 3 and 8 h post reperfusion after tMCAO. **b, c** Quantitative analysis of mean infarct volume and mean % hemispheric swelling of mouse brains 24 h post tMCAO. Data are means ± SEM, male mice, $n = 14$ (vehicle), 8 (Closantel 0.1), 8 (Closantel 1.0), 5 (Closantel 2.5), and 4 (WNK463 2.5). ***$p < 0.001$ and **$p < 0.01$ versus vehicle control, one-way ANOVA.

levels after IVH, and antagonizes IVH-induced phosphorylation of SPAK, NKCC1, and multiple KCCs in CPe. These data corroborate previous findings[1,3], and suggest ZT-1a or related strategies as potential nonsurgical treatment modalities for hydrocephalus. Further work will be required to determine the mechanisms of NKCC1 ion transport at the CPe membrane, and to assess the therapeutic potential of ZT-1a in other preclinical models of hydrocephalus.

In conclusion, we have developed ZT-1a, a novel drug that potently and selectively inhibits SPAK kinase, the master regulator of CCCs. ZT-1a inhibits NKCC1 and stimulates the KCCs by decreasing their regulatory phosphorylation. Intracerebroventricular delivery of ZT-1a, by decreasing inflammation-induced phosphorylation of CCCs in the choroid plexus, reduces CSF hypersecretion in hemorrhagic hydrocephalus. Systemically administered ZT-1a attenuates cerebral edema, protects against ischemic brain damage, and improves post-ischemic neurological outcomes by reducing ischemia-induced CCC phosphorylation. These results suggest that ZT-1a or related compounds are effective CCC modulators, with therapeutic potential for treatment of disorders of dysregulated brain volume homeostasis.

## Methods

**Reagents and general methods.** Protein sample buffer and bicinchoninic acid assay reagent were from Thermo Scientific (Rockford, IL). Closantel was from Sigma (St. Louis, MO) and WNK463 was from MedChemExpress (Princeton, NJ). STOCK1S-50699 and STOCK1S-14279 were from InterBioScreen (Chernogolovka, Russian Federation). Horseradish peroxidase (HRP)-conjugated anti-rabbit Ig was from Molecular Probes (Eugene, OR). RIPA buffer and enhanced chemiluminescence agent (ECL) reagent were from Pierce (Rockford, IL).

**Antibodies.** The Division of Signal Transduction Therapy Unit at the University of Dundee (RRID:SCR_011633) supplied antibodies to KCC3 (1 μg/ml, S701C), KCC3A phosphoT991 (1 μg/ml, S959C), KCC3A phosphoT1048 (1 μg/ml, S961C), NKCC1 (1 μg/ml, S022D), NKCC1 phospho-T203/T207/T212 (1 μg/ml, S763B), WNK1 (1 μg/ml, S079B), SPAK (1 μg/ml, S551D), OSR1 (1 μg/ml, S850C), SPAK/OSR1 (S-motif) phosphoS373/S325 (1 μg/ml, S670B), and full-length human ERK1 (1 μg/ml, S221B). The pan-KCC2 antibody (NeuroMab clone N1/12) was from NeuroMab. The FLAG M2 antibody (1 μg/ml, F3165) was from Sigma. The KCC4 antibody was from Novus Biologicals (0.4 μg/ml, NBP1-85133). Antibodies to GSK-3β phosphoS9 (1:1000 dilution, #9336), GAPDH (0.1 μg/ml, #2118), and β-Actin (1:5000 dilution, 8H10D10) were from Cell Signaling Technology. Anti-pGSK-3α/β (1:1000 dilution, #OPA1-03083) was from Thermo Scientific. Anti-Na$^+$–K$^+$–ATPase alpha subunit (1:1000 dilution, #a5c) was from the Developmental Studies Hybridoma Bank (RRID:SCR_013527, Iowa City, IA). Horseradish peroxidase-coupled secondary antibodies for immunoblotting were from Pierce. IgG used in control immunoprecipitation experiments was affinity-purified from pre-immune sera using Protein-G-Sepharose.

**Cell culture, transfections, and stimulations.** HEK-293 (human embryonic kidney 293) cells were cultured on 10-cm-diameter dishes in DMEM supplemented with 10% (v/v) fetal bovine serum, 2 mM L-glutamine, 100 U/ml penicillin, and 0.1 mg/ml streptomycin. HEK-293 cells were transfected with a mixture of 20 μl of 1 mg/ml polyethylenimine (Polysciences) and 5–10 μg of plasmid DNA as described previously[26]. Thirty-six hours post transfection, cells were stimulated with either control isotonic or hypotonic medium for a period of 30 min. Cells were lysed in 0.3 ml of ice-cold lysis buffer/dish, lysates were clarified by centrifugation at 4 °C for 15 min at 26,000 g, and the supernatant aliquots were frozen in liquid nitrogen and stored at –20 °C. Protein concentrations were determined using the Bradford method. Cells were treated with the indicated concentrations of the SPAK/OSR1 inhibitors. Isotonic buffer was 135 mM NaCl, 5 mM KCl, 0.5 mM CaCl$_2$, 0.5 mM MgCl$_2$, 0.5 mM Na$_2$HPO$_4$, 0.5 mM Na$_2$SO$_4$, and 15 mM HEPES (pH 7.5). Hypotonic low-chloride buffer was 67.5 mM sodium gluconate, 2.5 mM potassium gluconate, 0.25 mM CaCl$_2$, 0.25 mM MgCl$_2$, 0.5 mM Na$_2$HPO$_4$, 0.5 mM Na$_2$SO$_4$, and 7.5 mM HEPES (pH 7.5).

**Primary mouse neuronal cultures.** Embryonic day 14–18 wild-type pregnant mice, in a C57BL6J background, were deeply anesthetized with 5% isoflurane and euthanized by cervical dislocation[39]. Fetuses were removed from the uterus and the brain cortical tissues were dissected in ice-cold Hank's balanced salt solution. The cortical tissues were exposed to 0.125 mg/mL of trypsin at 37 °C for 20 min. The dissociated cells were centrifuged at 587 × g for 5 min at room temperature. The cells were gently mixed into a single-cell suspension before plating. Cells (200–1000 cells/mm²) were cultured on glass coverslips coated with poly-D-lysine in neurobasal medium containing B-27 supplements, L-glutamine, and penicillin/streptomycin (100 U/mL and 0.1 mg/mL, respectively). Cultures were incubated at 37 °C in an incubator with 5% CO$_2$ and atmospheric air. Neurons in culture for 7–9 days were used in the study.

**Human fibroblasts.** Primary fibroblast cell culture was established as described[88]. Passaged cells ($2 \times 10^5$ cells per coverslip) were cultured on glass coverslips coated with poly-D-lysine in Dulbecco's Modified Eagle Medium (DMEM) supplemented with 10% fetal bovine serum (FBS) and 0.1 mg/mL penicillin/streptomycin. Cultures were incubated at 37 °C in an incubator with 5% CO$_2$ and atmospheric air. Passages 9–21 were used in the study.

**$^{86}$Rb$^+$ uptake assay for KCC3 or NKCC1 activity.** $^{86}$Rb$^+$ uptake assays were performed on HEK-293 cells transfected with WT or mutant KCC3 plasmid DNAs as detailed in several of our previous publications[25,26]. For measurement of NKCC1 activity, HEK-293 cells were transiently transfected with empty vector or wt NKCC1 plasmid DNA. HEK-293 cells were plated at 50–60% confluence in 12-well plates (2.4-cm-diameter wells) and transfected with wild-type or various mutant forms of full-length flag-tagged human KCC3 (1 μg of plasmid DNA per well) in the presence of 2.5 μl of polyethylenimine (1 mg/ml). The $^{86}$Rb$^+$-uptake assay was performed on cells at 36 h post transfection. Culture medium was aspirated from the wells and replaced with either isotonic or hypotonic medium for 15 min at 37 °C, and then for a further 15 min with stimulating medium containing additional 1 mM ouabain (Oua, Na$^+$/K$^+$–ATPase inhibitor) and 0.1 mM bumetanide (Bum, inhibitor of NKCC1 cotransporter). For measurement of NKCC1 activity in HEK-293 cells, stimulating medium contained ouabain, and with or without further addition of bumetanide as for controls. This stimulating medium was then removed and replaced with isotonic medium containing inhibitors plus 2 μCi/ml $^{86}$Rb$^+$. After incubation for 10 min at 37 °C, cells were rapidly washed three times with the respective ice-cold nonradioactive medium. Washed cells were lysed in 300 μl of ice-cold lysis buffer and $^{86}$Rb$^+$ uptake was quantitated by liquid scintillation counting (PerkinElmer).

**Cell volume measurements.** Cell volume change was determined using Calcein-AM as a marker of intracellular water volume[89]. The coverslip-plated cells were

incubated with 0.5 μM calcein-AM for 30 min at 37 °C in the dark. The coverslip was then mounted in a heated (37 °C) imaging/perfusion chamber (Warner Instruments, Hamden, CT) on a Nikon Ti Eclipse inverted epifluorescence microscope equipped with 40× Super Fluor oil immersion objective, and a Princeton Instruments MicroMax CCD camera. Calcein fluorescence was monitored using a FITC filter set (excitation 480 nm, emission 535 nm, Chroma Technology, Rockingham, VT). Images were collected every 60 s with MetaFluor image-acquisition software (Molecular Devices, Sunnyvale, CA) and regions of interest (~20–25 cells) were selected. Baseline drift resulting from photobleaching and dye leakage was corrected as described before[90]. Fluorescence change was plotted as a function of the reciprocal of the relative osmotic pressure and the resulting calibration curve applied to all subsequent experiments as described before[90]. The HEPES-buffered isotonic solution contained (in mM, pH 7.4) 100 NaCl, 5.4 KCl, 1.3 CaCl$_2$, 0.8 MgSO$_4$, 20 HEPES, 5.5 glucose, 0.4 NaHCO$_3$, and 70 sucrose with osmometer-confirmed osmolarity of 310 mOsm (Advanced Instruments, Norwood, MA). Anisosmotic solutions (250, 370, 400, and 515 mOsm) were prepared by removal or addition of sucrose to the above solution.

**Immunoblot and phospho-antibody immunoprecipitation.** Whole-cell or cerebrospinal fluid (CSF) lysates were prepared with RIPA lysis buffer as previously described[3,91]. Lysate protein samples were subjected to immunoblot and immunoprecipitation as previously described[25]. Protein samples (40 μg) were boiled in sample buffer for 5 min, resolved by 7.5% sodium dodecyl sulfate polyacrylamide-gel electrophoresis, and electrotransferred onto a polyvinylidene difluoride membrane[25,92,93]. Membranes were incubated for 30 min with TBST (Tris-buffered saline, 0.05% Tween-20) containing 5% (w/v) skim milk. Blots were then washed six times with TBST and incubated for 1 h at room temperature with secondary HRP-conjugated antibodies diluted 5000-fold in 5% (w/v) skim milk in TBST. After repeating the washing steps, signals were detected with enhanced chemiluminescence reagent. Antibodies used in Supplementary Fig. 11 (anti-phospho-NKCC1, pNKCC1), anti-pSPAK/pOSR1, and antibody to total SPAK/OSR1 (tSPAK/tOSR1) were the kind gifts of Dr. Yang (Taiwan National University)[94]. Immunoblots were developed using a film automatic processor (SRX-101, Konica Minolta Medical) and films were scanned at 600 dpi (PowerLook 1000, UMAX). Figures were generated using Photoshop/Illustrator (Adobe). Band densities were measured with ImageJ. For phospho-antibody immunoprecipitation, KCC isoforms were immunoprecipitated from indicated cell extracts. Two mg of the indicated clarified cell extract was mixed with 15 μg of the indicated phospho-specific KCC antibody conjugated to 15 μl of protein-G-Sepharose, in the added presence of 20 μg of the dephosphorylated form of the phosphopeptide antigen, and incubated for 2 h at 4 °C with gentle shaking. Immunoprecipitates were washed three times with 1 ml of lysis buffer containing 0.15 M NaCl and twice with 1 ml of buffer A. Bound proteins were eluted with 1× LDS sample buffer.

**Model of post-hemorrhagic hydrocephalus.** All animal experiments were approved by the Institutional Animal Care and Use Committee (IACUC) of the Yale University, and in accordance with the guidelines and regulations in the NIH Guide for the Care and Use of Laboratory Animals. Male Wistar rats (Harlan, Indianapolis, IN, USA), age 8 weeks (220–230 g), were anesthetized (60 mg/kg ketamine plus 7.5 mg/kg xylazine, IP) and allowed to breathe room air spontaneously. Body temperature was maintained at 37 ± 1 °C (Harvard Apparatus, Holliston, MA, USA) throughout the course of the experiments. PHH was modeled using a modified protocol based on previously described methods[91,95]. Briefly, the tail artery of an anesthetized rat was aseptically cannulated using a flexible catheter (PE-20) preloaded with heparinized saline. The rat was then mounted in a stereotactic apparatus (Stoelting Co., Wood Dale, IL), a midline scalp incision was made to expose the skull, and a 1-mm burr hole was made using a high-speed drill over the right lateral ventricle (coordinates, $x = -0.8$, $y = -1.7$ mm relative to bregma). Approximately 200 μL of blood was drawn from the tail artery catheter and loaded into a 500-μL syringe (Hamilton, Reno, NV), which was mounted to the stereotactic frame. Under stereotactic guidance, 50 μL of freshly collected autologous blood, free from anticoagulants, was infused into the right lateral ventricle (coordinates, $x = -0.8$, $y = -1.7$, and $z = -4.5$ mm relative to bregma), over the course of 5 min, and the 26-gauge needle was held in place for an additional 20 min to prevent backflow of blood upon needle removal.

**Quantitation of rates of CSF production and intracerebroventricular drug administration.** Rates of CSF production were measured as previously described[91]. Briefly, anesthetized rats were mounted in a stereotactic apparatus and a 1.3-mm burr hole was made over the left lateral ventricle (coordinates, $x = -0.8$, $y = +1.7$ relative to bregma). Over the right lateral ventricle, an Alzet brain infusion cannula (#1; Durect, Cupertino, CA) was mounted with one spacer to adjust to −4.5-mm depth. The cannula was then connected to a 1-mL syringe loaded with either ZT-1a or vehicle solution (see below) via PE-20 catheter tubing to a syringe infusion apparatus (Pump elite 11, Harvard Apparatus). Next, the rat's head was rotated on the ear bars 90°, nose down, and the suboccipital muscles were dissected to the cisterna magna to expose the atlanto-occipital ligament. The ligament was punctured and a 23-gauge flexible catheter (PE-20) was advanced 5 mm through the foramen of Magendie to the 4th ventricle. Sterile, molecular-grade mineral oil

(100 μL, Sigma Aldrich, St. Louis, MO) was infused into the 4th ventricle to occlude the aqueduct of Sylvius, thereby creating a closed system of CSF circulation. With the rat in the same position, a glass capillary tube (cat # CV8010-300, borosilicate, OD, 1 mm; ID, 0.8 mm; length, 30 cm; VitroCom, Mountain Lakes, NJ) was advanced through the burr hole into the left lateral ventricle. The total volume (V) of CSF at a given time point was calculated as $V(mm^3) = \pi \cdot r^2 \cdot d$, where r is the radius of the capillary tube and d is the distance that CSF traveled within the capillary. The rate of CSF formation (μL/min) was calculated from the slope of the volume–time relationship. A baseline rate of spontaneous CSF secretion (no drug infusion) was calculated over 30 min prior to drug infusion, and then compared with the calculated rate of CSF formation following 30 min of ZT-1a infusion.

ZT-1a solution preparation. Intraventricular infusion solutions were made using sterile artificial cerebrospinal fluid (aCSF), composed of (in mM) sodium, 150; potassium, 3.0; calcium, 1.4; magnesium, 0.8; phosphorus, 1.0; chloride, 155, pH 7.19 (Tocris, Bristol, UK), with a calculated osmolarity of 311.2 mOsmC/l[3,91]. ZT-1a was dissolved in aCSF using DMSO as a co-solvent and titrated to pH 9 (10 mmol, 1% DMSO, pH 9). As a control, a vehicle solution of aCSF (1% DMSO, pH 9) omitting ZT-1a was prepared and tested.

**Tissue harvest and choroid plexus isolation.** Rats were euthanized via overdose of pentobarbital (Euthasol) administered IP and transcardially perfused with ice-cold normal saline. The brain was rapidly isolated and then placed in an ice-cold saline bath, after which the choroid plexus was carefully dissected under magnification using sharp forceps. Approximately 3 mg of choroid plexus tissue was harvested from each brain, collected into a 1.5-mL tube, and flash frozen in liquid nitrogen for storage.

**Middle cerebral artery occlusion model.** Focal cerebral ischemia was induced by transient occlusion of the left middle cerebral artery (MCA) for 50 min[42,92]. Under an operating microscope, the left common carotid artery was exposed through a midline incision. Two branches of the external carotid artery (ECA), occipital and superior thyroid arteries, were isolated and coagulated. The ECA was dissected further distally and permanently ligated. The internal carotid artery (ICA) was isolated and carefully separated from the adjacent vagus nerve. A 12-mm length of silicon-coated nylon filament (size 6-0, native diameter 0.11 mm; diameter with coating 0.21 ± 0.02 mm; coating length 5–6 mm; Doccol Corporation, Sharon, MA) was introduced into the ECA lumen through a puncture. The silk suture around the ECA stump was tightened around the intraluminal nylon suture to prevent bleeding. The nylon suture was then gently advanced from the ECA to the ICA lumen until mild resistance was felt (~9 mm). For reperfusion, the suture was withdrawn 50 min after MCAO to restore blood flow. Body temperature was maintained for the duration of the experiment between 36.5 °C and 37 °C with a small-animal heating pad (Kent Scientific).

**Preparation of brain membrane and cytosolic protein fractions and immunoblotting.** Brain homogenates were prepared at 24 h post reperfusion[92]. Mice were anesthetized with 5% isoflurane vaporized in N$_2$O and O$_2$ (3:2), then decapitated. The contralateral (CL) and ipsilateral (IL) brain tissues were dissected in five volumes of cold homogenization buffer (0.25 M sucrose, 10 mM Tris-HCl, 1 mM EDTA, pH 7.4, protease, and phosphatase inhibitor cocktail, Pierce). Brain tissues were gently homogenized with a tissue grinder (Kontes, Vineland, NJ, USA) for 10 strokes in homogenization buffer. The homogenized samples were centrifuged at $1000 \times g$ at 4 °C for 10 min. The supernatant (S1) was collected and centrifuged at $\sim 200,000 \times g$ for 30 min in a Beckman Optima$^{TM}$ XL-80k Ultracentrifuge. The cytosolic fraction [supernatant (S2)] and crude membrane pellet were collected. The pellet was resuspended in the homogenization buffer. Protein content in both membrane and cytosolic fractions was determined by the standard bicinchoninic acid method.

**Kinome profiling.** Kinome profiling was performed using KinomeScan ScanMAX at 1 μM compound concentrations (see Supplemental Data). Protocols are available from DiscoverX (https://www.discoverx.com/).

**Animal preparation.** All animal experiments were approved by the University of Pittsburgh Institutional Animal Care and Use Committee and performed in accordance with the National Institutes of Health Guide for the Care and Use of Laboratory Animals. Nine- to 14-week-old C57BL/6 J mice (both male and female) were used in the study (Jackson laboratories, Bar Harbor, ME).

**Drug treatment.** Vehicle 100% DMSO (2 ml/kg body weight/day), WNK463 (2.5 mg/kg body weight/day), Closantel (0.1–2.5 mg/kg body weight/day,) or ZT-1a (2.5–5.0 mg/kg body weight/day) were administered via intraperitoneal injection (i.p., Fig. 7a), with an initial half-dose at 3 h and the second half-dose at 8 h following reperfusion.

**Cerebral blood flow measurement.** Regional cerebral blood flow was measured using a two-dimensional laser speckle contrast analysis system (PeriCam PSI High Resolution with PIMSoft, Perimed)[65]. Isoflurane-anesthetized mice were

head-fixed using stereotaxic equipment during imaging. The skin was retracted to expose the intact skull. Images were taken at 19 frames/second with averaging. Average signal intensity was taken from fixed-size (0.5-mm$^2$) regions of interest drawn over the parietal bone plate on the ipsi (IL)- and contralateral (CL) sides. Percent perfusion values were taken in comparison with the mean values of the pre-ischemic ipsilateral side. Five consecutive images at each time period per animal were averaged for analysis.

**Brain infarction volume and hemispheric swelling measurements**. At 24 h of reperfusion, mice were anesthetized with 5% isoflurane and then decapitated. Coronal brain tissue slices (2 mm) were stained for 20 min at 37 °C with 1% 2,3,5-triphenyltetrazolium chloride monohydrate (TTC, Sigma, St. Louis, MO) in PBS solution. Infarction volume was calculated with correction for edema using ImageJ software as described[96]. The extent of hemispheric swelling was calculated using the following equation: swelling (% contralateral side) = [(volume of ipsilateral hemisphere − volume of contralateral hemisphere)/volume of contralateral hemisphere] × 100[92].

**Neurological deficit score**. A neurological deficit grading system[42,92] was used to evaluate neurological deficit at 0, 1, 3, 5, and 7 days after tMCAO. The scores are 0, no observable deficit; 1, forelimb flexion; 2, forelimb flexion and decreased resistance to lateral push; 3, forelimb flexion, decreased resistance to lateral push, and unilateral circling; 4, forelimb flexion and partial or complete lack of ambulation.

**Corner test**. Corner test was used to assess MCAO-induced sensorimotor abnormalities as described previously[42]. In brief, the corner test apparatus consists of two cardboards (30 × 20 × 1 cm each) placed together at a 30° angle to form a narrow alley. The mouse was placed between the two angled boards facing the corner. When exiting the corner, uninjured mice turned left or right randomly. After tMCAO, animals with unilateral brain damage displayed an asymmetry in corner turning. The numbers of left and right turns of each mouse during 10 trials were recorded, and turning movements that were not part of a rearing movement were not scored. Preoperative training was carried out twice per day for 3 days prior to operation. Postoperatively, animals were tested on days 1, 3, 5, and 7.

**Adhesive-removal test**. An adhesive-removal test was used to measure somato-sensory deficits as described previously[42,65]. In brief, two small pieces of adhesive tape (4 × 3 mm) were attached to the forepaws in an alternating sequence and with equal pressure by the experimenter before each trial. Animals were released into a testing cage, and the time of contact and removal of the adhesive patch were recorded. Contact was recorded when either shaking of the paw or mouth contact occurred. The trial ended after the adhesive patch was removed or after 2 min had elapsed. Preoperative training was carried out twice per day for 3 days prior to operation. Postoperatively, animals were tested on days 1, 3, 5, and 7.

**DTI ex vivo brains**. Seven days post reperfusion, mice were anesthetized with 5% isoflurane, transcardially perfused with 4% paraformaldehyde (PFA), and decapitated[42]. For ex vivo MRI, brains were maintained within the skull to avoid anatomical deformation. After overnight postfixation in 4% PFA, heads were stored in PBS solution at 4 °C. MRI was performed at 500 MHz using a Bruker AV3HD 11.7 T/89-mm vertical-bore small-animal MRI scanner, equipped with a 25-mm quadrature RF coil and Paravision 6.0 (Bruker Biospin, Billerica, MA). Following positioning and pilot scans, a DTI data set covering the entire brain was collected using a multislice spin echo sequence with 3 reference and 30 noncollinear diffusion-weighted images with the following parameters: TE/TR = 22/5000 ms, 4 averages, matrix size = 192 × 192 reconstructed to 256 × 256, field of view = 17.3 × 17.3 mm, 33 axial slices, slice thickness = 0.5 mm, $b$ value = 1200 s/mm$^2$, and Δ/δ = 10/5 ms.

DTI data were analyzed with DSI Studio (http://dsistudio.labsolver.org/)[42]. In a blinded manner, region of interests (ROIs) were manually drawn for the ipsilateral (IL) and contralateral (CL) cortex, striatum, corpus callosum (CC), and external capsule (EC). Values for the fractional anisotropy (FA), afferent diffusion coefficient (ADC), and diffusivity were calculated for the entire volume of each ROI. For atrophy assessment during the subacute phase (7 days post stroke), percentage hemispheric shrinkage (% shrinkage) was determined by the difference in volume between two hemispheres and then divided by the CL hemispheric volume according to the formula: % Shrinkage = (CL volume − IL volume)/CL volume × 100, as described[97,98].

**Statistical analysis**. Animal subjects were randomly assigned into different studies and surgical procedures, and investigators blinded to experimental conditions performed data analyses. The number of animals studied was 80% powered to detect 20% changes with α (2-sided) = 0.05. A total of 168 mice were used in the study, and no results were excluded from the analysis. Data were expressed as means ± SEM. Statistical significance was determined by student's t test, or one-way or two-way ANOVA using the Tukey's post hoc test in the case of multiple comparisons (GraphPad Prism 7.0, San Diego, CA, USA). Neurological deficit scores were analyzed by the nonparametric Mann–Whitney test. A probability

value < 0.05 was considered statistically significant. Detailed methods are available in the Supplementary Informaiton.

**Reporting summary**. Further information on research design is available in the Nature Research Reporting Summary linked to this article.

## Data availability
The data that support the findings of this study are available from the corresponding author upon reasonable request. The source data underlying Figs. 2b, 3a, b, 4b, c, 5c, 6c, e, f, 7b–d, 8b, and 9b, c and Supplementary Figs. 2, 3, 4b, d, 5–8, 9c, 10b, c, e, f, 11b, c, and 12 are provided as a Source Data file.

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

## Acknowledgements

We thank D. R. Alessi (Dundee) for his support. This work was supported by grants from the National Key R&D Program and the National Natural Science Foundation of China (No. 2017YFA0504504, 2016YFA0502001, 91853203, and 81661138005 to X.D.) and (No. 81970238 to J.Z.), the Fundamental Research Funds for the Central Universities of China (No. 20720190101 to X.D.), NIH Grants R01 NS38118 (D.S.) and R01NS109358 (K.K.), the Hydrocephalus Foundation and Simons Foundation (K.K.), and VA I01BX002891-01A1 (D.S.).

## Author contributions

Conceived and designed the experiments: J.Z., M.H.B., K.K., D.S. and X.D. Performed the experiments: J.Z., M.H.B., T.Z., J.K.K., Z.W., V.M.P., J-F.Z., H.H., M.N.H., W.H., L.M.F., T.K.H. and B.J.M. Analyzed the data: J.Z., M.H.B., T.Z., J.K.K., Z.W., M.B.M., S.M.P., X.D., K.K. and D.S. Contributed reagents/materials/analysis tools: D.D., M.M., A.E.S., A.J.T. and R.P. Wrote the paper: J.Z., M.H.B., S.L.A., K.K., D.S. and X.D.

## Competing interests

The authors declare no competing interests.
