## [Peer Review File · Nature Communications]

Reviewers' Comments:

Reviewer #1:

Remarks to the Author:

General Comments

This manuscript presents an ambitious study that was undertaken to evaluate a newly developed SPAK kinase inhibitor, ZT-1a, for its effectiveness in NKCC1 inhibition, KCC stimulation, reduction of CSF hypersecretion in post-hemorrhagic hydrocephalus and attenuation of ischemic stroke induced cerebral infarct and neurological damage. The results presented describing a broad range of experiments suggest that ZT-1a is an effective kinase-cation chloride cotransport modulator with therapeutic potential for brain disorders of water and ion dysregulation. The findings presented are timely and hold strong potential for significant impact in the field. However, the manuscript is extremely dense and somewhat poorly organized and presented such that much of the important findings are not easily appreciated. There are many more supplementary figures than figures presented within the main manuscript. Without the information from the supplementary figures the main manuscript loses a lot of important information. Why not work to include more of the supplementary figures in the main manuscript? I have a number of specific concerns and suggestions for revision that I provide in comments to the authors. A number of these relate to misstatements and overinterpretation of the data.

Specific Comments

1. Title: The title should be revised to better reflect the findings in the study. This is because there are no experiments that evaluated restoration of water homeostasis.
2. Abstract; In line 9 from the top, the statement that "ZT-1a promotes reduced cellular ion influx and stimulates net cellular Cl extrusion..." is misleading because no experiments evaluating Cl were conducted in this study. Please clarify that the findings are consistent with the hypothesis that ZT-1a will have the stated effect rather than implying that the study demonstrated Cl efflux.
3. Page 3, Introduction: In the top paragraph, line 8 and also bottom paragraph lines 3-4; it is shortsighted not to mention that blood brain barrier endothelial cell NKCC also participates in brain edema formation during ischemic stroke and that ischemic conditions increase phosphorylation and activity of NKCC in these cells. (e.g., O'Donnell et al *Journal of Cerebral Blood Flow and Metabolism* 24: 1046-1056, 2004; Foroutan et al *AJP Cell* 289: C1492-C1501, 2005; Kawai et al, *J. Neurochem* 66:2572-2579, 1996; Kawai et al *Neurochem Research* 21: 1259-1266, 1996; Spatz et al, *AJP Cell* 272: C231-C239, 1997; O'Donnell et al, *Brain Edema: From Molecular Mechanisms to Clinical Practice*, J. Badaut and N. Plesnila, eds., San Diego:Academic Press, Chapter 7, pp 129-150, 2017;). ZT-1a will certainly target NKCC (and KCC) where it is found in other cells of the neurovascular unit. This must be at least mentioned.

Also, in the top paragraph of page 4, lines 9-11, why not mention tPA treatment as well here?
4. Page 4, second paragraph: Please clarify here that the cells used in these studies were HEK cells. Also, as in the Abstract, the statement the ZT-1a stimulated cellular Cl extrusion needs to be revised since Cl extrusion was not assessed. Also, the statement that ZT-1a normalizes CSF hypersecretion by decreasing SPAK-mediated phosphorylation of NKCC1/KCC4 needs to be revised. It is more correct to state that the findings are consistent with...or support the hypothesis, etc.
5. Page 4, Results; In the first paragraph of Results, please give just a bit more information about the "scaffold-hybrid" strategy. Figure 1A alone is not adequate to explain the approach.
6. Figure 1 and legend: Use of hypotonic low Cl to activate SPAK and increase phosphorylation of NKCC1 and KCC3 needs to be explained more fully. NKCC1 is well known to be stimulated by hypertonicity and reduced intracellular Cl while it is inhibited by hypotonicity. Using hypotonic low

Cl solutions to activate SPAK in the HEK cells is a good experimental tool but it is clearly not physiological. One would hope to see that ZT-1a would have its desired coordinate effect on reducing NKCC and increasing KCC activity in a more relevant cell type and pathophysiological setting. Please address this and clarify reasoning. It is important to put these findings in context and avoid over-extrapolating results.

In Figure 1 and through the remainder of the manuscript, why was the focus almost entirely on KCC? For example, in Figure 1C, why not also show quantitated data for p-NKCC1? If the goal of the study is to show that ZT-1a effectively reduces phosphorylation of both NKCC and KCC to decrease and increase their activities, respectively, why the primary focus on KCC?

7. Page 4, Results, Figure 1B and Supplementary Table 1: The table needs more explanation. Are the values shown IC50s? Also, in the 4th-5th lines from the bottom of the page "ZT-1a emerged as the most potent compound, inhibiting phosphorylation of NKCC1...by 72%. Where are these data shown? Only representative Westerns of p-NKCC1 are presented.

8. Page 5, Results: Top paragraph and Table 2, data in the table show that CLK2, GSK3b, MAPKAP2-K2, PKA, PRAK and ULK2 were all reduced at least 50% by 10 μ M Zt-1a and that GSK3b was also reduced by just 1 μ M ZT-1a. How do these data support ZT-1a high selectivity for SPAK? Please clarify.

9. Page 5, Results, second full paragraph and Figure S4-6: Please include information about the two peptides in the legends and main manuscript text. In particular, it would be helpful to specifically state why the two different motif-containing peptides were used and the significance of the results. This can be gleaned after study Methods, and the figure but it's difficult for the reader to keep straight what the experiments are that are being conducted and their significance without going back and forth between main text and Supplemental figures, legends and Methods.

10. Page 5, Results, bottom line: The section title "ZT-1a promotes KCC3-dependent cellular Cl extrusion: needs to be changed. No measurements of Cl extrusion were performed.

11. Page 6, Results, bottom paragraph: The title of this section should be revised. I suggest that "pathological CSF hypersecretion" be changed to just "CSF hypersecretion". Using both pathological and hypersecretion is redundant, is it not? Also, it would be better to change "by decreasing NKCC1/KCC4 phosphorylation" to "and decreases NKCC1/KCC4 phosphorylation". These experiments don't demonstrate that the effect of ZT-1a is via decreasing phosphorylation of the cotransporters, only that both occur, suggesting that ZT-1a effects on the cotransporters underlies the reduction of CSF secretion.

12. Page 7, Results, first full paragraph and Figure S7A: In line 7, "we examined ZT-1a penetration in ischemic brains with damaged, leaky BBB" is presumptive. Unless BBB integrity was assessed one can only assume that the BBB was, in fact, leaky under the imposed conditions. Also "damaged" is a broad term. The BBB undergoes many changes during ischemia and subsequent reperfusion, not all of them can be considered "damage". It is surprising that "compared to Sham controls, a 6-fold increase in plasma ZT-1a levels" were observed at 2 hrs. Does this mean that while ZT-1a was injected into both Sham-operated and tMCAO animals, 2 hours later the plasma ZT-1a was 6-fold higher than in the non-ischemic animals? As written, the sentence is confusing. Also, what is the explanation for the increase in plasma ZT-1a? One would not expect that to occur.

13. Page 8 Results, last paragraph: Figure 6A-C purportedly shows that ZT-1a more effectively reduced ischemic brain injury than Clonsantel. This is not apparent, however. Figure 3 shows that ZT-1a at 5 mg/kg reduced infarct and swelling by ~ 50%. Figure 6 shows that Clonsantel at 2.5 mg/kg reduced infarct and swelling by ~50%. Please clarify this issue.

14. Statistics: Specific information regarding both n values and statistical tests need to be provided for each figure and table.

15. Page 9, Discussion: In the first line of this page (continuing from the bottom of page 8), the statement that ZT-1a normalizes CSF hypersecretion...by decreasing inflammation-dependent up-regulated phosphorylation of SPAK, NKCC1 and KCC4. This overstates what was observed. Inflammation nor inflammation-dependent upregulated phosphorylation of SAPK etc was not assessed. Also, stating that ZT-1a "restores brain water homeostasis" is over-reaching. What is the basis of that statement? It reduces infarct and swelling but water homeostasis is a very broad term.

In the second paragraph on this page "We have circumvented problems previously associated with targeting of NKCC of NCCs by focusing on the upstream SPAK kinase...". What problems are referred to here? In the last line of the second paragraph, regarding retinal toxicity of Closantel, what is known about whether ZT-1a has any retinal effects?

In the third paragraph of the Discussion, please do not repeat what was presented in Results (including, e.g. References to Figure 1A and Table S1). In the last line of this paragraph, "...effectively promoting cell Cl extrusion." Again, one can't make conclusions about Cl extrusion when it wasn't assessed.

16. Page 10, Discussion: in the top two lines "...reduced brain atrophy at 7 days post stroke". Where are the brain atrophy data? In the second paragraph "ZT-1a more effectively entered the Ischemic brain across its leaky BBB". Again, no assessment of BBB leakiness was made. Further, one can't conclude that the compound simply crossed a leaky barrier. It is possible that the increased permeability occurred via a vesicular pathway. In line 5 of this paragraph, "facilitating brain access for small molecule drugs" should be "...for many small molecule drugs". BBB permeability depends not only on molecular size but also charge, lipophilicity, whether it is transported by the BBB "drug" transporters, etc.

17. Page 20 and Methods in Supplemental Information: The ⁸⁶Rb assay needs to be better explained.

...bumetanide, to prevent ⁸⁶Rb uptake via the NKCC1 cotransporter...". Was the experiment designed to assess NKCC activity by media with ouabain vs ouabain + bumetanide? Or was the experiment designed to assess ⁸⁶Rb uptake with both Na/K pump and NKCC inhibited with the assumption that remaining flux was KCC? If it is assumed that all remaining ⁸⁶Rb uptake is via KCC, what about K channel "leak flux"?

Reviewer #2:

Remarks to the Author:

This manuscript by Zhang et al describes a novel inhibitor targeting the SPAK kinase and subsequent phosphorylation of NKCC1/KCCs. The manuscript addresses inclusion of this inhibitor in settings of CSF hypersecretion and brain edema formation. The manuscript is interesting and the inhibitor may be highly useful in the clinic with the challenges associated with brain water disturbances. With the superficial method description and very brief mention in the result section it is a little challenging to evaluate the technical soundness and the reasoning behind some of the experimental choices (see below). The manuscript could thus benefit from a little more attention to detail (and reasoning for conducting the different experiments and for choosing the KCC isoforms) and possibly a restructuring (see below) to aid the reader along.

Specific points:

Introduction: "CCC subtypes include the Cl--importing, Na+- driven CCCs (NCC, NKCC1, and NKCC2; the "N[K]CCs"), and the Cl--exporting, K+-driven CCCs (KCC1-4; the "KCCs)". The direction of the transport is determined by the ion gradient in a given cell type, not by its name.

NKCC1 is obviously outwardly transporting in CPE (although not in many other cells), as also recognized by the authors' earlier work. Please correct. And again further down in the introduction. Although it may well be the case that CCCs are strongly involved in volume regulation in the mammalian brain, it was not my impression that it has been solidly demonstrated (especially given the ongoing argument as to whether NKCC1 is even expressed in astrocytes and endothelium, when these are not cultured). Please verify and/or correct.

No quantification is carried out on Fig. 1B, please include. Fig. 1C why is there no activation of KCC activity in hypotonic medium? Why include hypotonic medium if there is no effect of it? Suppl fig 7. In text it says i.p. injection, in Fig. legend s.c. injection? Is SPAK not an intracellular kinase? Why would the inhibitor not be able to exit the fenestrated endothelium in the choroid plexus and make it into the CPe?

Not all studies find evidence of KCC expressed in the CPE luminal membrane. Please provide some form of evidence that the employed KCC4 antibody is specific – and it really is KCC4 you are looking at (not that much in CP according to RNAseq analysis, more KCC1). With IP of the active site (?), how do you know what you pull down without verification of AB specificity? KCCs are then supposedly active with dephosphorylation. If you believe that it resides in the luminal membrane (along side NKCC1?) then how does that align with the reduced CSF formation? It is stated in the discussion again that KCC4 is involved in the PHH – I do not recall the evidence of it being the case (did the authors not earlier assign the increased CSF formation as bumetanide-sensitive)? Whether or not KCC4 is phosphorylated? And why only test KCC4 in CP?

N's are sometimes included in the text, sometimes not.

Fig. 4. This gel is a little more messy. It looks like the inhibitor causes a phosphorylation of SPAK in the contralateral side? Perhaps also true for KCCs. Is that the case? Was that expected? How many times were these gels run? I see no mention of that anywhere? Hopefully not just once with n=3 animals? It would be great with an 'n' in the text in association with the p-values. The text regarding this figure is pretty short and does not fully explain the many panels included. Why all the relations to the Na/K-pump without mentioning why (in the result text). It comes across a little confusing even though the take home is probably straight forward. Why no explanation of the choice of KCC3? We hear a lot about KCC2 in neurons – not about KCC3 – is it even there? How did you verify that your Ab was specific towards that isoform. Inhibition of an ion-extruder should make the cells swell more, one would think, so why is it only better to use this inhibitor? Again mentioned in the discussion – why is KCC3 the more important one?

With Fig. 5 showing essentially the same as Fig. 3, it is odd that they are presented apart. One has a sense of the WB being the carrying structure here, and take up so much space that much of the other data have been put in supplementary. I would recommend structuring it a little differently so that one question is CSF formation, one is edema formation/neuronal survival and one is the p-WB assigning the effect to the SPAK/CCCs – or something like that. As it stands it comes, unnecessarily, across as trying a bunch of different things instead of presenting the tight story that it actually is. It certainly is convenient to just run one gel with all three animals in but one could make it much clearer with running of one sample and showing that in the figure (+the quantification) – and just having all the big gels in supplementary (if possible).

Discussion: "ZT-1a promotes cellular Cl⁻ extrusion by simultaneous reduction of SPAK-dependent NKCC1 stimulatory phosphorylation and KCC inhibitory phosphorylation. Intracerebroventricular delivery of ZT-1a, by decreasing inflammation-induced NKCC1/KCC4 phosphorylation in the choroid plexus, normalizes CSF hypersecretion in hemorrhagic hydrocephalus" – again does not align with NKCC1 direction in CPe and the authors' own data?

Figure legends sometimes mention the statistical test, sometimes not. Sometimes SDs are given, sometimes SEMs why?

Methods are very superficial (which I believe is not in line with nature communication guidelines). How do you get Rb⁺ influx of an inwardly-directed transporter? I am sure there are tricks to that but to not even mention it is not favorable.

Why all the immunoprecipitation of the KCCs? I do not see that explained and do not understand why they have to be treated in that manner? Except that some researchers generally find very little KCC4 (and 3) in CPE both on protein level, functionally, and mRNA – are they simply just detectable if you do IP first? Or why? Again, it is crucial to describe why the authors have chosen

KCC3 and KCC4 for neurons and CPe, when these are not the most expressed in these cell types?

Reviewer #3:

Remarks to the Author:

The manuscript by Kahle and colleagues describes the discovery of a SPAK and OSR1 kinases inhibitors that restores brain water homeostasis in vivo. The work, overall, is of great quality and offers a validation of previous genetic work made by Dandan Sun in this area. The authors conducted and presented a series of experiments in this work that went from the design and synthesis of the molecules and their in vitro/in vivo properties. The in vitro and in vivo characterisation of the compound, ZT-1a, was very detailed while that of the design of the molecule is too brief and lacks critical details. This makes the manuscript in this format rather imbalanced. To address this, I have the following comments:

1. There was lack of details on how the compounds were designed. What properties were the authors aiming to achieve when designing SPAK and OSR1 kinase inhibitors, BBB penetration? The authors discussed that in their mouse model the BBB was leaky. Can you they comment on whether in humans this is also the case?
2. The compound studied in this work, ZT-1a, is of high lipophilicity ($\log P > 6$). In the in vivo studies, it was solubilised and administered in 100% DMSO as a vehicle. The authors need to provide the volume of DMSO used in each treatment dosing and that in the control. The property of the molecule being very lipophilic needs to be included in the manuscript.
3. Minor comment: the structure of Rafoxanide is not shown in Figure 1A. It needs to be included.

After addressing the imbalance in the manuscript, I think it could be suitable for publication in nature communications.

We thank the reviewers for their critical review and helpful suggestions. We have revised the manuscript accordingly. In addressing reviewers' specific concerns, we conducted new experiments and included new data in the revision. We believe that the revised manuscript has been significantly improved and the conclusion is further strengthened with the new positive findings. Following are our point-by-point responses.

Reviewer #1 (Remarks to the Author):

General Comments:

This manuscript presents an ambitious study that was undertaken to evaluate a newly developed SPAK kinase inhibitor, ZT-1a, for its effectiveness in NKCC1 inhibition, KCC stimulation, reduction of CSF hypersecretion in post-haemorrhagic hydrocephalus and attenuation of ischemic stroke induced cerebral infarct and neurological damage. The results presented describing a broad range of experiments suggest that ZT-1a is an effective kinase-cation chloride cotransporter modulator with therapeutic potential for brain disorders of water and ion dysregulation. The findings presented are timely and hold strong potential for significant impact in the field. However, the manuscript is extremely dense and somewhat poorly organized and presented such that much of the important findings are not easily appreciated. There are many more supplementary figures than figures presented within the main manuscript. Without the information from the supplementary figures the main manuscript loses a lot of important information. Why not work to include more of the supplementary figures in the main manuscript? I have a number of specific concerns and suggestions for revision that I provide in comments to the authors. A number of these relate to misstatements and over interpretation of the data.

Response: We have now moved key findings from the Supplemental section into the main manuscript.

Specific Comments

Q1: Title: The title should be revised to better reflect the findings in the study. This is because there are no experiments that evaluated restoration of water homeostasis.

Response: We have revised the title; the new title is "Therapeutic modulation of brain cation-Cl⁻ cotransport via the novel SPAK kinase inhibitor ZT-1a"

Q2: Abstract; in line 9 from the top, the statement that "ZT-1a promotes reduced cellular ion influx and stimulates net cellular Cl extrusion..." is misleading because no experiments evaluating Cl were conducted in this study. Please clarify that the findings are consistent with the hypothesis that ZT-1a will have the stated effect rather than implying that the study demonstrated Cl efflux.

Response: We have deleted this sentence in the revised abstract.

Q3: Page 3, Introduction: In the top paragraph, line 8 and also bottom paragraph lines 3-4; it is short-sighted not to mention that blood brain barrier endothelial cell NKCC also participates in brain edema formation during ischemic stroke and that ischemic conditions increase phosphorylation and activity of NKCC in these cells. (e.g., O'Donnell et al Journal of Cerebral Blood Flow and Metabolism 24: 1046-1056, 2004; Foroutan et al AJP Cell 289: C1492-C1501, 2005; Kawai et al, J. Neurochem 66:2572-2579, 1996; Kawai et al Neurochem Research 21: 1259-1266, 1996; Spatz et al, AJP Cell 272: C231-C239, 1997; O'Donnell et al, Brain Edema: From Molecular Mechanisms to Clinical Practice, J. Badaut and N. Plesnila, eds., San Diego:Academic Press, Chapter 7, pp 129-150, 2017;). ZT-1a will certainly target NKCC (and KCC) where it is found in other cells of the neurovascular unit. This must be at least mentioned.

Response: We apologize for not including in our original description of stroke damage the significance of NKCC1 activity in the blood-brain barrier (BBB) endothelial cells and in the neurovascular unit. As recommended by the reviewer, we have revised the following sentences in the Introduction section (Page 3, top paragraph, line 8-12): "Coordinated transmembrane influx and efflux of ions and water are also necessary for cell volume homeostasis in neurons, glia, and blood-brain barrier (BBB) endothelial cells. Impaired cell volume regulation following ischemic stroke and other brain injuries can lead to cytotoxic cell swelling, disruption of BBB integrity and cerebral edema (Kahle et al., 2009; Simard et al., 2007; O'Donnell et al., 2017)."

Page 4, 2nd paragraph, lines 7-11: "SPAK-regulated, CCC-mediated ion transport has been implicated in the pathogenesis of multiple brain pathologies associated with impaired brain ion and water homeostasis. Experimental ischemic cerebral edema is associated with increased phosphorylation of the SPAK/OSR1 T-loop and NKCC1 (Thr203/Thr207/Thr212) in both neurons and oligodendrocytes (Begum et al., 2015), and in BBB endothelial cells (Kawai et al., 1996; O'Donnell et al., 2004)."

Q4: Also, in the top paragraph of page 4, lines 9-11, why not mention tPA treatment as well here?

Response: As suggested by Reviewer #1, we have revised the top paragraph on page 3 (lines 17-20) to include tPA and thrombectomy: "Recent advances in vascular stroke therapy such as clot lysis by recombinant tissue plasminogen activator (rtPA) and clot removal by radiologically guided thrombectomy are appropriate for fewer than 8% of ischemic stroke patients (Kleindorfer et al., 2013; Smith et al., 2017)."

Q5: Page 4, second paragraph: Please clarify here that the cells used in these studies were HEK cells. Also, as in the Abstract, the statement the ZT-1a stimulated cellular Cl⁻ extrusion needs to be revised since Cl⁻ extrusion was not assessed. Also, the statement that ZT-1a normalizes CSF hypersecretion by decreasing SPAK-mediated phosphorylation of NKCC1/KCC4 needs to be revised. It is more correct to state that the findings are consistent with...or support the hypothesis, etc.

Response: on page 4 and 5, we have indicated that measurements of Cl⁻-dependent K⁺ efflux (KCC3 or NKCC1 activity) sensitive to ZT-1a were conducted in HEK293 cells transiently expressing KCC (Figures 1 D and S2). We also removed the phrase "cellular Cl⁻ extrusion" from the Abstract and on page 5.

Q6: Page 4, Results; In the first paragraph of Results, please give just a bit more information about the "scaffold-hybrid" strategy. Figure 1A alone is not adequate to explain the approach.

Response: Page 4, in the first paragraph of Results "Scaffold-hybrid" strategy means that we select the pharmacophores, the key structural units of the desired bioactivity, from the multitude of different scaffolds. We then design new compounds through combining these pharmacophores in a novel manner with the goal of achieving better bioactivity. In this work, we designed new compounds through the combination of pharmacophores of the 2-(4-amino-2-chloro-5-methylphenyl)-2-(4-chlorophenyl)acetonitrile moiety from Closantel and the chloro-substituted 2-hydroxybenzoic acid from STOCK1S-14279 and Rafxoxanide (Figure 1A). We have clarified the sentence as below: "This "scaffold-hybrid" strategy, by extracting and combining the pharmacophores from different scaffolds, has proven useful for development of highly selective kinase inhibitors (Deng et al., 2011; Deng et al., 2013)."

Q7: Figure 1 and legend: Use of hypotonic low Cl to activate SPAK and increase phosphorylation of NKCC1 and KCC3 needs to be explained more fully. NKCC1 is well known to be stimulated by hypertonicity and reduced intracellular Cl while it is inhibited by hypotonicity. Using hypotonic low Cl solutions to activate SPAK in the HEK cells is a good experimental tool but it is clearly not physiological. One would hope to see that ZT-1a would have its desired coordinate effect on reducing NKCC and increasing KCC activity in a more relevant cell type and pathophysiological setting. Please address this and clarify reasoning. It is important to put these findings in context and avoid over-extrapolating results.

Response: We agree with Reviewer #1's critiques. We have conducted experiments to directly assess cell volume regulation in primary mouse neurons and human fibroblasts in response to hypertonicity-mediated cell shrinkage. Newly added Figure 3 shows that SPAK inhibitor ZT-1a accelerated hypertonicity-mediated cell shrinkage of cultured primary neurons. The following paragraph has been included on page 8, lines 4-16: "SPAK-NKCC1 signaling plays an important role in maintaining cell volume homeostasis. Effects of SPAK

inhibitor ZT-1a on cell volume regulation were assessed in primary cortical neurons exposed to hypertonic (370 mOsm/kg H₂O) osmotic stress (Figure 3A). As shown in Figure 3B, hypertonic osmotic stress led to 45% cell volume decrease with an initial rate of 24 ± 5 % vol/min. In the presence of bumetanide (BMT) or ZT-1a, neurons exhibited severe cell shrinkage (86 % and 87 %, respectively, $p < 0.005$, $n = 4-5$, Figure 3B), although only bumetanide increased initial rate of shrinkage (45 ± 4 % vol/min and 29 ± 9 % vol/min). These similar effects of BMT and ZT-1a treatment suggest targeting of the same signaling cascade. Upon returning to isotonic conditions after 20 min exposure to hyperosmotic stress, neuronal cell volume recovery was rapid, at rates indistinguishable among groups (Figure 3C). This phenomenon was also observed in human fibroblast cells (Figure S9)."

Q8: In Figure 1 and through the remainder of the manuscript, why was the focus almost entirely on KCC? For example, in Figure 1C, why not also show quantitated data for p-NKCC1? If the goal of the study is to show that ZT-1a effectively reduces phosphorylation of both NKCC and KCC to decrease and increase their activities, respectively, why the primary focus on KCC?

Response: Figure 1B Western blots were subjected to quantitative densitometry, as shown in revised Figure 1C and Figure S2A, including p-KCCs and p-NKCC1. We have also conducted the ⁸⁶Rb⁺ uptake assays for Cl⁻-dependent K⁺ flux (NKCC1 activity) in response to treatment with ZT-1a (Figure S2B). We revised the sentence in the main text (page 6, lines 42-46): "...suggesting that Ala mutation of these Thr residues abolishes SPAK-mediated inhibitory phosphorylation. Further, ZT-1a (from 1 μ M) significantly inhibited NKCC1-mediated K⁺ extrusion in isotonic or hypotonic low Cl⁻ conditions ($p < 0.05$; $n = 3$; Figure S2B). These results show that ZT-1a stimulates KCC3 activity by decreasing its SPAK-dependent inhibitory phosphorylation at Thr991/Thr1048 and inhibits NKCC1 activity."

Q9: Page 4, Results, Figure 1B and Supplementary Table 1: The table needs more explanation. Are the values shown IC50s? Also, in the 4th-5th lines from the bottom of the page "ZT-1a emerged as the most potent compound, inhibiting phosphorylation of NKCC1...by 72%. Where are these data shown? Only representative Westerns of p-NKCC1 are presented.

Response: Values presented in Supplementary Table 1 are based on the quantitative densitometry data of Figure 1B-C, Figure S1 and Figure S2A.

Q10: Page 5, Results: Top paragraph and Table 2, data in the table show that CLK2, GSK3b, MAPKAP2-K2, PKA, PRAK and ULK2 were all reduced at least 50% by 10 μ M Zt-1a and that GSK3b was also reduced by just 1 μ M ZT-1a. How do these data support ZT-1a high selectivity for SPAK? Please clarify.

Response: Page 5, lines 29-29, ZT-1a inhibits SPAK kinase in a non-ATP-competitive manner (Figure S4) by disrupting the binding interaction between WNK1 and SPAK (Figure S5). ZT-1a potently inhibits p-NKCC1 or p-KCC from concentrations as low as 1-3 μ M (Figure 1B). Whereas KCC3 Thr991/Thr1048 phosphorylation was significantly inhibited at 1 and 3 μ M ZT-1a, GSK3beta Ser9 phosphorylation in HEK-293 cells was unaffected by 3 μ M ZT-1a (Figure S4A-B). We also examined ZT-1a effects on the inhibition of multiple additional kinases in ischemic brains, and detected no marked inhibition of p-JNK, p-ERK or p-GSK-3alpha/beta (Figure S4C-D).

Q11: Page 5, Results, second full paragraph and Figure S4-6: Please include information about the two peptides in the legends and main manuscript text. In particular, it would be helpful to specifically state why the two different motif-containing peptides were used and the significance of the results. This can be gleaned after study Methods, and the figure but it's difficult for the reader to keep straight what the experiments are that are being conducted and their significance without going back and forth between main text and Supplemental figures, legends and Methods.

Response: In order to further establish that ZT-1a binds to the CCT domains of SPAK and OSR1, we subjected lysates of transfected HEK-293 cells to a SPAK antibody pull-down assay in the presence of varying concentrations of ZT-1a or of human WNK4-derived RFQV 18-mer peptide. AFQV 18mer peptide, known not to bind the primary pocket of SPAK and OSR1 (Villa et al., 2007), served as negative control. We also tested the effects of varying concentrations of the inhibitors STOCK1S-50699 and Closantel. Therefore, on page 5, lines 42-45; page 6, lines 1-2, we have revised the paragraph from line 42 as follows: "In order to establish that ZT-1a binds to SPAK and OSR1 CCT domains, we subjected lysates of transfected HEK293 cells to a SPAK antibody pull-down assay in the presence of increasing ZT-1a concentrations, and

compared results with those from increasing concentrations of human WNK4 RFQV 18-mer peptide (Villa et al., 2007) and with AFQV 18mer negative control peptide, which does not bind the primary pocket of SPAK and OSR1 (Villa et al., 2007).”

Q12: Page 5, Results, bottom line: The section title “ZT-1a promotes KCC3-dependent cellular Cl extrusion: needs to be changed. No measurements of Cl extrusion were performed.

Response: we agree with the Reviewer’s comments. We revised the section title on page 6 as “ZT-1a inhibits NKCC1 but stimulates KCC3 activity”. KCCs transport potassium (K⁺) and chloride (Cl⁻) ions out of the cell with 1:1 stoichiometry. Therefore, we measure the K⁺ flux mediated by KCC3 activity or NKCC1 activity in HEK263 cells overexpressing either protein in response to the ZT-1a treatment.

Q13: Page 6, Results, bottom paragraph: The title of this section should be revised. I suggest that “pathological CSF hypersecretion” be changed to just “CSF hypersecretion”. Using both pathological and hypersecretion is redundant, is it not? Also, it would be better to change “by decreasing NKCC1/KCC4 phosphorylation” to “and decreases NKCC1/KCC4 phosphorylation”. These experiments don’t demonstrate that the effect of ZT-1a is via decreasing phosphorylation of the cotransporters, only that both occur, suggesting that ZT-1a effects on the cotransporters underlies the reduction of CSF secretion.

Response: As suggested, we have revised the sentence on page 7 as “Intracerebroventricular (ICV) delivery of ZT-1a reduced CSF hypersecretion and up-regulated CCC phosphorylation in the choroid plexus.”

Q14: Page 7, Results, first full paragraph and Figure S7A: In line 7, “we examined ZT-1a penetration in ischemic brains with damaged, leaky BBB” is presumptive. Unless BBB integrity was assessed one can only assume that the BBB was, in fact, leaky under the imposed conditions. Also “damaged” is a broad term. The BBB undergoes many changes during ischemia and subsequent reperfusion, not all of them can be considered “damage”. It is surprising that “compared to Sham controls, a 6-fold increase in plasma ZT-1a levels” were observed at 2 hrs. Does this mean that while ZT-1a was injected into both Sham-operated and tMCAO animals, 2 hours later the plasma ZT-1a was 6-fold higher than in the non-ischemic animals? As written, the sentence is confusing. Also, what is the explanation for the increase in plasma ZT-1a? One would not expect that to occur.

Response: We agree with Reviewer #1 about the need to demonstrate altered permeability of the BBB after 50-min tMCAO in mice. In the revised manuscript, we have included new data (Figure 4A-C) assessing serum-albumin extravasation into brain parenchyma, an assay to evaluate the BBB integrity in brain tissues. -Page 8, par. 2 lines 21-25 have been revised as follows: “First, we examined intrinsic immunofluorescence labelling of serum-albumin at 24 h after tMCAO as an indication of BBB integrity. As shown in Figure 4A-C, serum albumin extravasation into brain parenchyma was ~5-fold higher in mice with ischemic stroke than in sham-operated mice (p < 0.05, n = 5-6), suggesting that tMCAO/reperfusion induced disruption of BBB integrity in ischemic brains”

-Page 11, lines 35-41 have been revised as follows: “Plasma half-life of ZT-1a is only ~1.8 hrs in normal naïve mice, and ZT-1a penetration across the healthy BBB is minimal. Ischemic stroke injury disrupts BBB tight junctions as early as 30 min after resumption of perfusion (Sandoval and Witt, 2008; Shi et al., 2016), facilitating brain access for many small molecule drugs (Won et al., 2011). We have shown that ischemic stroke caused robust extravasation of serum albumin into brain parenchyma in parallel with detection of elevated ZT-1a levels in ischemic brain. These findings suggest that ZT-1a more effectively entered ischemic brain parenchyma across the leaky BBB.”

-“ZT-1a plasma bioavailability in ischemic mice”: To address Reviewer #1’s concerns on elevated ZT-1a plasma bioavailability in ischemic mice, we performed new experiments with additional mice, which reduced variance of the data. The following sentences on page 8, lines 26-30 have been revised with these new results as: “Two hours after ZT-1a administration (5 mg/kg; i.p.), plasma ZT-1a levels in ischemic and non-ischemic sham mice were indistinguishable (p = 0.29; n = 10-11; Figure 4E), whereas ZT-1a concentration in both CL and IL hemispheres of ischemic brains were ~ 1.8-fold higher than in sham brains (p = 0.006; n = 10-11; Figure 4F).”

Q15: Page 8 Results, last paragraph: Figure 6A-C purportedly shows that ZT-1a more effectively reduced ischemic brain injury than Clonsantel. This is not apparent, however. Figure 3 shows that ZT-1a at 5 mg/kg reduced infarct and swelling by ~ 50%. Figure 6 shows that Clonsantel at 2.5 mg/kg reduced infarct and swelling by ~50%. Please clarify this issue.

Response: It is correct that comparable neuroprotective effects were detected in mice treated with the novel SPAK inhibitor ZT-1a or with antiparasitic drug Closantel at similar doses 2.5-5 mg/kg. However, at ~25 mg/kg, Closantel shows adverse effects in human (Essabar et al., 2014; Tabatabaei et al., 2016). To address this issue, we conducted crude toxicity testing of ZT-1a or Closantel at 25 mg/kg dose in normal or ischemic stroke mice (Supplemental Fig. S12). No mortality was detected in ZT-1a- or Closantel-treated naïve mice. However, ischemic stroke mice treated with Closantel exhibited 100% mortality. In contrast, ZT-1a-treated mice displayed significantly prolonged overall survival. Additional pharmacokinetics and toxicity studies of ZT-1a in post-stroke brains are required to determine safe and optimal effective doses of ZT-1a. We have included this new result as Supplemental Figure S12 and in Results page 10, lines 9-10, as “Moreover, toxicity of Closantel and ZT-1a at 25 mg/kg in naïve and ischemic stroke mice was analysed with Kaplan-Meier survival curves (Figure S12). Overall survival of Ischemic stroke mice treated with ZT-1a was significantly prolonged compared to that of Closantel-treated mice ($p = 0.008$, $n = 6$, Log-rank test).”

Q16: Statistics: Specific information regarding both n values and statistical tests need to be provided for each figure and table.

Response: Specific information regarding both n values and statistical tests is now provided for each figure and table.

Q17: Page 9, Discussion: In the first line of this page (continuing from the bottom of page 8), the statement that ZT-1a normalizes CSF hypersecretion...by decreasing inflammation-dependent up-regulated phosphorylation of SPAK, NKCC1 and KCC4. This overstates what was observed. Inflammation nor inflammation-dependent up-regulated phosphorylation of SAPK etc was not assessed. Also, stating that ZT-1a “restores brain water homeostasis” is over-reaching. What is the basis of that statement? It reduces infarct and swelling but water homeostasis is a very broad term.

Response: On page 12, lines 15-25, we have added discussions for addressing these points: “The CPe secretes higher volumes of fluid (CSF) per cell than any other epithelium (~500 ml/day). NKCC1 expressed in the apical CPe contributes approximately half of CSF production. However, the mechanism by which NKCC1 contributes to CSF production remains unclear, in view of NKCC1’s unique apical localization in the CPe compared to its basolateral position in all other epithelia (Steffensen et al., 2018). Indeed, some have proposed inward NKCC1-mediated flux to be required for apical K^+ recycling and continued CSF production (Delpire and Gagnon, 2018; Gregoriades et al., 2019), while others have provided evidence for net ion efflux with obligatory water transport to directly contribute to CSF production (Karimy et al., 2017; Steffensen et al., 2018). Nonetheless, in a rat model of haemorrhagic hydrocephalus, intraventricular haemorrhage causes a Toll-like receptor 4 (TLR4)- and NF- κ B-dependent inflammatory response in CPe associated with ~3-fold increase in bumetanide-sensitive CSF secretion (Karimy et al., 2017).”

Q18: In the second paragraph on this page “We have circumvented problems previously associated with targeting of NKCC of NCCs by focusing on the upstream SPAK kinase...”. What problems are referred to here? In the last line of the second paragraph, regarding retinal toxicity of Closantel, what is known about whether ZT-1a has any retinal effects?

Response: Please see responses for Q15.

Q19: In the third paragraph of the Discussion, please do not repeat what was presented in Results (including, e.g. References to Figure 1A and Table S1). In the last line of this paragraph, “.... effectively promoting cell Cl extrusion.” Again, one can’t make conclusions about Cl extrusion when it wasn’t assessed.

Response: We have deleted the references to Figure 1A and Table S1. We have also removed the wording “cell Cl extrusion.” And changed to “....effectively promoting cellular Cl^- -dependent K^+ efflux”.

Q20: Page 10, Discussion: in the top two lines “...reduced brain atrophy at 7 days post stroke”. Where are the brain atrophy data? In the second paragraph “ZT-1a more effectively entered the Ischemic brain across its leaky BBB”. Again, no assessment of BBB leakiness was made. Further, one can’t conclude that the compound simply crossed a leaky barrier. It is possible that the increased permeability occurred via a vesicular pathway. In line 5 of this paragraph, “facilitating brain access for small molecule drugs” should be

“...for many small molecule drugs”. BBB permeability depends not only on molecular size but also charge, lipophilicity, whether it is transported by the BBB “drug” transporters, etc.

Response: “Brain atrophy”: Figure 5D (right panel) illustrates % hemisphere shrinkage, which reflects brain atrophy. The descriptions are included on page 9, lines 16-18: “T2-weighted MRI analysis further confirmed that ZT-1a treatment reduced stroke-induced lesion volume by ~40% and brain atrophy (hemisphere shrinkage) by ~41% ($p < 0.05$, $n = 6$, Figure 5D)”. Moreover, we have added the detailed method of brain atrophy analysis in Methods (page 29, lines 44-46, and page 30, lines 1-6).

“BBB permeability of ZT-1a and BBB drug transporters in ischemic brains” : please see our responses to Q14.

“Brain access of drugs”: as suggested by Reviewer #1, the sentence on page 11, lines 35-41 has been revised as: “Ischemic stroke injury increases permeability and disrupts BBB tight junctions as early as 30 minutes after resumption of perfusion (Sandoval and Witt, 2008; Shi et al., 2016), facilitating brain access for many small molecule drugs (Won et al., 2011)”.

Q21: Page 20 and Methods in Supplemental Information: The ^{86}Rb assay needs to be better explained. ...bumetanide, to prevent ^{86}Rb uptake via the NKCC1 cotransporter...”. Was the experiment designed to assess NKCC activity by media with ouabain vs. ouabain + bumetanide? Or was the experiment designed to assess ^{86}Rb uptake with both Na/K pump and NKCC inhibited with the assumption that remaining flux was KCC? If it is assumed that all remaining ^{86}Rb uptake is via KCC, what about K channel “leak flux”?

Response: We have included in Methods (page 25) more detailed information regarding our $^{86}\text{Rb}^+$ -uptake experiments: “The $^{86}\text{Rb}^+$ uptake assay was performed on HEK293 cells transfected with wt or mutants KCC3 plasmid DNA as described previously. For measurement of NKCC1 activity, HEK293 cells were transiently transfected with empty vector or wt NKCC1 plasmid DNA. HEK293 cells were plated at 50–60% confluence in 12-well plates (2.4-cm-diameter per/well) and transfected with wild type or various mutant forms of full-length flag-tagged human KCC3. Each well of HEK293 cells was transfected with 2.5 μl of 1 mg/ml polyethylenimine and 1 μg of plasmid DNA. The $^{86}\text{Rb}^+$ -uptake assay was performed on cells at 36 hours post-transfection. Culture medium was aspirated from the wells and replaced with either isotonic or hypotonic medium for 15 min at 37°C, then for a further 15 min with stimulating medium containing additional 1 mM ouabain (Oua, Na⁺/K⁺-ATPase inhibitor) and 0.1 mM bumetanide (Bum, inhibitor of NKCC1 cotransporter). For measurement of NKCC1 activity in HEK293 cells, stimulating medium contained ouabain or ouabain plus bumetanide. This stimulating medium was then removed and replaced with isotonic medium containing inhibitors plus 2 $\mu\text{Ci/ml}$ $^{86}\text{Rb}^+$. After incubation for 10 min at 37°C, cells were rapidly washed three times with the respective ice-cold non-radioactive medium. Washed cells were lysed in 300 μl of ice-cold lysis buffer and $^{86}\text{Rb}^+$ uptake was quantitated by liquid scintillation counting (PerkinElmer).” We transiently over-expressed NKCC1 for our $^{86}\text{Rb}^+$ uptake assay to provide a large NKCC1 signal over the background K channel “leak flux”.

Reviewer #2 (Remarks to the Author):

This manuscript by Zhang et al describes a novel inhibitor targeting the SPAK kinase and subsequent phosphorylation of NKCC1/KCCs. The manuscript addresses inclusion of this inhibitor in settings of CSF hypersecretion and brain edema formation. The manuscript is interesting and the inhibitor may be highly useful in the clinic with the challenges associated with brain water disturbances. With the superficial method description and very brief mention in the result section it is a little challenging to evaluate the technical soundness and the reasoning behind some of the experimental choices (see below). The manuscript could thus benefit from a little more attention to detail (and reasoning for conducting the different experiments and for choosing the KCC isoforms) and possibly a restructuring (see below) to aid the reader along.

Specific points:

Q22: Introduction: “CCC subtypes include the Cl⁻-importing, Na⁺- driven CCCs (NCC, NKCC1, and NKCC2; the “N[K]CCs”), and the Cl⁻-exporting, K⁺-driven CCCs (KCC1-4; the “KCCs”)”. The direction of the transport is determined by the ion gradient in a given cell type, not by its name. NKCC1 is obviously outwardly transporting in CPE (although not in many other cells), as also recognized by the authors’ earlier work. Please correct. And again further down in the introduction.

Response: On page 3, lines 27-32, We have corrected this as follows: “In epithelial cells under most physiological conditions (with possible exception of choroid plexus) the Na⁺-driven CCCs NCC, NKCC1, and NKCC2 (“N(KCCs”), Dowd and Forbush, 2003; Moriguchi et al., 2005; Piechotta et al., 2003; Piechotta et al., 2002; Richardson et al., 2008; Richardson et al., 2011), function as Cl⁻ importers, whereas the Na⁺-independent KCC1-4 (“KCCs, Adragna et al., 2004; Arroyo et al., 2013; de Los Heros et al., 2014; Zhang et al., 2016), function as Cl⁻ exporters.”.

Q23: Although it may well be the case that CCCs are strongly involved in volume regulation in the mammalian brain, it was not my impression that it has been solidly demonstrated (especially given the ongoing argument as to whether NKCC1 is even expressed in astrocytes and endothelium, when these are not cultured). Please verify and/or correct.

Response: We believe the following statement to be well supported by collective evidence to date. “The coordinated regulation of CCC function is important for cell volume regulation in most brain cells, preventing excessive cell swelling or shrinkage in response to osmotic or ischemic stress (Kahle et al., 2015; Zhang et al., 2016). The central importance of CCCs to CSF homeostasis by choroid plexus has been recently recognized (Karimy et al., 2017; Steffensen et al., 2018).” With regard to the reviewer’s concerns that some of these data were obtained from cultured cells, we note that several unbiased studies illustrate expression of NKCC1 mRNA and protein in astrocytes in vivo. Two recent RNA-seq studies of acutely purified brain cells by the late Ben Barres at Stanford (<http://www.brainrnaseq.org/>) and by the Betscholtz lab at Karolinska (<http://betsholtzlab.org/VascularSingleCells/database.html>) detected NKCC1 (slc12a2) transcripts expressed in mouse brain astrocytes, and endothelial cells, as well as in neurons and oligodendrocytes (Zhang et al. 2014, *J Neurosci.*; Vanlandewijck et al. 2018, *Nature*). Moreover, we have detected NKCC1 protein in GFAP⁺ astrocytes and NSE⁺ neurons in adult rat brain (Yan et al. 2001, *Brain Research*; Zhang, , et al., 2014).

Zhang et al. (2014). An RNA-Sequencing Transcriptome and Splicing Database of Glia, Neurons, and Vascular Cells of the Cerebral Cortex. *J. Neurosci.*, 34 (36) 11929-11947.

Vanlandewijck M., et al. (2018). A molecular atlas of cell types and zonation in the brain vasculature. *Nature*, 554, 475-480.

Yan, Y., et al., (2001). Expression of Na⁺-K⁺-Cl⁻ cotransporter in rat brain during development and its localization in mature astrocytes. *Brain Research*, 911, 43–55.

Q24: No quantification is carried out on Fig. 1B, please include. Fig. 1.C why is there no activation of KCC activity in hypotonic medium? Why include hypotonic medium if there is no effect of it?

Response: Quantitative densitometry of Figure 1B Western blots was carried out and presented in Figure 1C and Figure S2A, including p-KCCs and p-NKCC1. We have also conducted ⁸⁶Rb⁺ uptake assays of NKCC1 function in the absence and presence of ZT-1a (Figure S2B).

Q25: Suppl fig 7. In text it says i.p. injection, in Fig. legend such injection? Is SPAK not an intracellular kinase? Why would the inhibitor not be able to exit the fenestrated endothelium in the choroid plexus and make it into the CPE?

Response: Thanks for noting this error, which we have corrected, noting intraperitoneal (i.p.) administration instead of subcutaneous injection.

According to Koumangoye and Delpire 2016 (*Am J Physiol Cell Physiol.* 2016 Jul 1;311(1):C43-53.), SPAK and OSR1 kinases entering cells through exosomes are preferentially expressed at the plasma membrane and that the kinases in exosomes are functional and maintain NKCC1 in a phosphorylated state.

Q26: Not all studies find evidence of KCC expressed in the CPE luminal membrane. Please provide some form of evidence that the employed KCC4 antibody is specific – and it really is KCC4 you are looking at (not that much in CP according to RNAseq analysis, more KCC1). With IP of the active site (?), how do you know what you pull down without verification of AB specificity? KCCs are then supposedly active with dephosphorylation. If you believe that it resides in the luminal membrane (alongside NKCC1?) then how does that align with the reduced CSF formation? It is stated in the discussion again that KCC4 is involved in the PHH – I do not recall the evidence of it being the case (did the authors not earlier assign the increased CSF formation as bumetanide-sensitive)? Whether or not KCC4 is phosphorylated? And why only test KCC4 in CP?

Response: Page 7, lines 17-26 now present the relevant references. “Choroid plexus epithelium (CPE) is the most actively secreting epithelium in the body (Damkier et al., 2013; Lehtinen et al., 2013), producing up to 500 cc/day of CSF (Cutler et al., 1968). SPAK kinase is expressed in CPE (Karimy et al., 2017), at higher levels than in any other tissue, including kidney (Zhang et al., 2015). Several CCCs are also expressed in choroid plexus (Karadsheh et al., 2004; Pearson et al., 2001; Praetorius and Nielsen, 2006) including NKCC1, recently shown essential for CSF secretion (Karimy et al., 2017; Steffensen et al., 2018). IVH-triggered TLR4 signaling stimulates CSF secretion > 3.5-fold and causes hydrocephalus by increasing functional expression of pSPAK and pNKCC1 in CPE (Karimy et al., 2017). We speculated that intracerebroventricular (ICV) delivery of ZT-1a into the cerebrospinal fluid might bypass blood-brain-barrier drug permeability issues and allow Zt-1a to exert its effects on SPAK and CCCs in CPE.”

To detect KCC1-4 proteins in rat choroid plexus, we first verified immuno-specificities of each antibody using flag-tagged KCC1-4 individually expressed in HEK293 cells, as demonstrated in Figure 2A. Site 2 phospho-KCC2 antibody pull-down fractions were subjected to SDS-PAGE followed by western blot with KCC isoform-specific antibodies (see Figure 2A for immunospecificity). Lysates from rat choroid plexus demonstrated robust expression of KCC1, KCC3 and KCC4, while KCC2 was below the detection limit in this tissue. We observed that phosphorylation of KCC1 and KCC3 at Sites-1 and -2 are significantly reduced in CPE, as we similarly showed for KCC4 phosphorylation at Sites-1 and -2.

Q27: Not all studies find evidence of KCC expressed in the CPE luminal membrane. Please provide some form of N's are sometimes included in the text, sometimes not.

Response: See Q26.

Q28: Fig. 4. This gel is a little more messy. It looks like the inhibitor causes a phosphorylation of SPAK in the contralateral side? Perhaps also true for KCCs. Is that the case? Was that expected? How many times were these gels run? I see no mention of that anywhere? Hopefully not just once with n=3 animals? It would be great with an 'n' in the text in association with the p-values. The text regarding this figure is pretty short and does not fully explain the many panels included. Why all the relations to the Na/K-pump without mentioning why (in the result text). It comes across a little confusing even though the take home is probably straightforward. Why no explanation of the choice of KCC3? We hear a lot about KCC2 in neurons – not about KCC3 – is it even there? How did you verify that your Ab was specific towards that isoform. Inhibition of an ion-extruder should make the cells swell more, one would think, so why is it only better to use this inhibitor? Again mentioned in the discussion – why is KCC3 the more important one?

Response: Na⁺-K⁺ ATPase α -subunit and n values: As crude membrane fractions from mouse ischemic brain tissues were used for Western blot analysis of SPAK, NKCC1 and KCC3, we chose to use Na/K-ATPase α -subunit as loading control for membrane protein. We have added this information to the legend of Fig. 6 (page 16, lines 12-13) as “Na⁺-K⁺ ATPase α -subunit served as loading control for membrane protein fraction”. We used 5 animals per group in this experiment, for which protein samples were tested at minimum in duplicate.

KCC2 and KCC3 protein expression in brains – although KCC2 has been considered the dominant neuronal isoform, Barres and colleagues detected similar abundances of KCC2/SLC12A5 and KCC3/SLC12A6 transcripts in acutely purified mouse brain neurons (<http://www.brainrnaseq.org/>) (Zhang et al. 2014, J Neurosci. 34, 11929-11947). In our study, we examined changes of NKCC1 and KCC3 protein in ischemic brain. We did not assess KCC2 protein in our current experiments, since we previously observed no significant change in KCC2 protein abundance in ischemic mouse brains (unpublished data). We have therefore added the following on page 9, line 38-41: “Since neurons express KCC2 as well as KCC3 (Zhang et al. 2014, J Neurosci) and KCC2 protein expression was unchanged in ischemic mouse brains (data not shown), we focused on assessing changes of NKCC1 and KCC3 protein in ischemic brain in this study.”

For specificity of antibody – please see our response to Reviewer #2 Q26.

Q29: With Fig. 5 showing essentially the same as Fig. 3, it is odd that they are presented apart. One has a sense of the WB being the carrying structure here, and take up so much space that much of the other data have been put in supplementary. I would recommend structuring it a little differently so that one question is CSF formation, one is edema formation/neuronal survival and one is the p-WB assigning the effect to the

SPAK/CCCs – or something like that. As it stands it comes, unnecessarily, across as trying a bunch of different things instead of presenting the tight story that it actually is. It certainly is convenient to just run one gel with all three animals in but one could make it much clearer with running of one sample and showing that in the figure (+the quantification) – and just having all the big gels in supplementary (if possible).

Response: as suggested by the reviewer, we have combined the original Fig. 3 and 5 to create a new, revised Fig. 5.

Q30: Discussion: “ZT-1a promotes cellular Cl⁻ extrusion by simultaneous reduction of SPAK-dependent NKCC1 stimulatory phosphorylation and KCC inhibitory phosphorylation. Intracerebroventricular delivery of ZT-1a, by decreasing inflammation-induced NKCC1/KCC4 phosphorylation in the choroid plexus, normalizes CSF hypersecretion in haemorrhagic hydrocephalus” – again does not align with NKCC1 direction in CPe and the authors’ own data?

Response: We have revised this sentence from the original:

“We have shown here that ICV administration of ZT-1a restores CSF secretion rates to basal levels after IVH and antagonizes inflammation-induced phosphorylation of SPAK, NKCC1 and KCC4 in CPe. These data suggest ZT-1a as a novel pharmacological treatment for hydrocephalus, a condition currently treatable only by the highly morbid surgical approaches of endoscopy or shunting.”

to the following revised version:

“We have shown here that ICV administration of ZT-1a reduces CSF secretion rates to basal levels after IVH and antagonizes IVH-induced phosphorylation of SPAK, NKCC1, KCC1, KCC3 and KCC4 in CPe. These data suggest pharmacological blockade of SPAK as a novel treatment for hydrocephalus, a condition currently treatable only by the highly morbid surgical approaches of endoscopy or shunting.”

Q31: Figure legends sometimes mention the statistical test, sometimes not. Sometimes SDs are given, sometimes SEMs why?

Response: We have repeated all the statistical analyses and uniformly presented SEMs.

Q31: Methods are very superficial (which I believe is not in line with nature communication guidelines). How do you get Rb⁺ influx of an inwardly directed transporter? I am sure there are tricks to that but to not even mention it is not favorable.

Response: Many of the detailed methods previously provided in the supplementary section have now been moved to the main text Methods section.

Q32: Why all the immunoprecipitation of the KCCs? I do not see that explained and do not understand why they have to be treated in that manner? Except that some researchers generally find very little KCC4 (and 3) in CPE both on protein level, functionally, and mRNA – are they simply just detectable if you do IP first? Or why? Again, it is crucial to describe why the authors have chosen KCC3 and KCC4 for neurons and CPe, when these are not the most expressed in these cell types?

Response: As noted in response to Q26, we have added “To detect KCC1-4 proteins in rat choroid plexus, we first verified immuno-specificities of each antibody using flag-tagged KCC1-4 individually expressed in HEK293 cells, as demonstrated in Figure 2A. Site 2 phospho-KCC2 antibody pull-down fractions were subjected to SDS-PAGE followed by western blot with KCC isoform-specific antibodies (see Figure 2A for immuno-specificity. Lysates from rat choroid plexus demonstrated robust expression of KCC1, KCC3 and KCC4, while KCC2 was below the detection limit in this tissue.”

Reviewer #3 (Remarks to the Author):

The manuscript by Kahle and colleagues describes the discovery of a SPAK and OSR1 kinases inhibitors that restores brain water homeostasis in vivo. The work, overall, is of great quality and offers a validation of previous genetic work made by Dandan Sun in this area. The authors conducted and presented a series of experiments in this work that went from the design and synthesis of the molecules and their in vitro/in vivo properties. The in vitro and in vivo characterisation of the compound, ZT-1a, was very detailed while that of

the design of the molecule is too brief and lacks critical details. This makes the manuscript in this format rather imbalanced. To address this, I have the following comments:

Q33: There was lack of details on how the compounds were designed. What properties were the authors aiming to achieve when designing SPAK and OSR1 kinase inhibitors, BBB penetration? The authors discussed that in their mouse model the BBB was leaky. Can you they comment on whether in humans this is also the case?

Response: The details of compound design were added in the answers to Q6 (Figure 1A). Our aim is to design new inhibitors with improved SPAK inhibitory activity and BBB penetration. To optimize BBB penetration, a multiparameter optimization algorithm for properties of central nervous system drugs (CNS MPO) was applied (Ref: Wager, T. T.; et al. ACS chemical neuroscience 2010, 1 (6), 435-49.). Higher CNS MPO scores suggest better BBB penetration. As shown in Figure Rx, ZT-1a has higher CNS MPO score than closantel, in part reflecting its lower molecular weight (Figure R1).

Figure R1. Comparison of CNS MPO score of Closantel with ZT-1a.

“ischemic stroke-mediated increase in BBB permeability”: please see revised Figure 4A and our responses to Reviewer #1 Q14. Our finding is consistent with results in ischemic human brain with disrupted BBB integrity (Krueger et al. 2017; Kassner et al 2015). Human post-stroke autopsy brain tissues show pronounced extravasation into brain parenchyma of blood components, including fibrinogen, albumin, as assessed by immunofluorescence analysis (Sladojevic et al, J Neurosci, v39, pp 743-757, 2019; Krueger et al, JCBFM, v37, pp 2539-2554, 2017). The following discussion has been added on page 12, lines 4-7: “Ischemic stroke-induced disruption of the BBB integrity is also evident in human brains (Kassner and Merali, 2015; Krueger et al., 2017), suggesting that small molecule neuroprotective drugs, such as ZT-1a, could potentially traverse the leaky BBB into ischemic human brain.”

Q34: The compound studied in this work, ZT-1a, is of high lipophilicity ($\log P > 6$). In the in vivo studies, it was solubilised and administered in 100% DMSO as a vehicle. The authors need to provide the volume of DMSO used in each treatment dosing and that in the control. The property of the molecule being very lipophilic needs to be included in the manuscript.

Response: On page 28, lines 30-33, the following sentences have been revised to include DMSO volume used in each treatment dosing: “Vehicle 100% DMSO (2 ml/kg body weight/day), WNK463 (2.5 mg/kg body weight/day), Closantel (0.1-2.5 mg/kg body weight/day) or ZT-1a (2.5-5.0 mg/kg body weight/day) were administered via intraperitoneal injection (i.p., Figure 5A), with an initial half-dose at 3 hr and the second half-dose at 8-hr following reperfusion”.

Q35: Minor comment: the structure of Rafoxanide is not shown in Figure 1A. It needs to be included.

Response: Thanks for suggestion. We have added the structure of Rafoxanide in Figure 1A.

After addressing the imbalance in the manuscript, I think it could be suitable for publication in Nature Communications.

Response: We thank the Reviewer 3 and the other Reviewers for their support of our manuscript.

Reviewers' Comments:

Reviewer #1:

Remarks to the Author:

The authors have been very responsive to my comments and suggestions for revision. This is now a much-improved manuscript. I have only one remaining issue. It relates to new data presented in Figure 3 of the revised manuscript. There is an inconsistency between the Results text and the legend regarding the level of hypertonicity used in the experiments. The legend indicates that it was 515 mOsm but the Results text indicates that 370 mOsm was used. Also, the authors state the ZT-1a accelerated the hypertonicity-mediated cell shrinkage. That does not make sense. When cells are exposed to hypertonic solutions they shrink, acting as a perfect (or near perfect) osmometer as water quickly moves out of the cell down its concentration gradient. The resulting cell shrinkage then stimulates NKCC activity to mediate a regulatory volume increase. This needs to be corrected. I have three other concerns about this figure and how it is presented and interpreted. The first is that if the cells were truly switched from 310 mOsm to 515 mOsm the cell volume would be expected to decrease to 58% of control volume. In the figure shown that is approximately what occurred in the control cells but volume decreased further in cells treated with bumetanide or ZT-1a. Do the authors have an explanation for that? Why would cells shrink further when NKCC is inhibited? Second, in the control cells the volume at first decreased rapidly as expected but then continued to decrease at a slower rate. I suspect that this could be an artifact of evaluating cell volume using Calcein. Third, the "volume recovery rate" that the authors refer to does not seem to be a true volume recovery, at least one not mediated by cellular mechanisms because the cells are simply being switched from hypertonic back to isotonic medium. It's important to clarify here what is true volume regulatory mechanism versus passive changes in volume due to altering the osmotic gradient across the cell membrane. Could the authors please address these issues?

Reviewer #2:

Remarks to the Author:

This revised manuscript by Zhang et al describes a novel inhibitor targeting the SPAK kinase and subsequent phosphorylation of NKCC1/KCCs. The manuscript addresses inclusion of this inhibitor in settings of CSF hypersecretion and brain edema formation. The revised manuscript is interesting and has addressed a few of the critique points raised in the last round. I, however, still find it difficult to navigate in some of the figures and the associated (brief) text, which are sometimes not provided in the same order. I still have some concerns with the manuscript as it stands and I (again) would urge the authors to consider re-writing parts of the manuscript to clarify some points and aid the reader in understanding the study.

Introduction: "Vectorial ion transport across apical and basolateral membranes of the choroid plexus epithelium (CPE), accompanied by osmotically obligated transport of water, results in daily cerebrospinal fluid (CSF) secretion of > 500 cc/day into brain ventricular spaces¹".

The particular study cited argued exactly against the point of osmotically obliged transport of water, so puzzling that the authors would cite that in this particular context?

"Impaired ionic homeostasis in CPE can result in hydrocephalus (accumulation of excess CSF in the brain ventricles), as in the settings of intraventricular hemorrhage (IVH) and infection^{2, 3}"

Has it ever been shown that it is the ion homeostasis in the CP epithelial cells that result in hydrocephalus? I looked through one of the cited articles but was unable to find that statement supported there? Please verify and/or rephrase – or cite the study by Karimy et al., which illustrates hyperactivity of an ion transporter as the etiology.

Again (see Q23 from last round): "These evolutionarily-conserved transporters²³, are of particular

importance in regulation of ion and water homeostasis in mammalian central nervous system (CNS)^{24, 25}. The coordinated regulation of CCC function is important for cell volume regulation in most brain cells, preventing excessive cell swelling or shrinkage in response to osmotic or ischemic stress^{22, 26}.”

I certainly do not disagree on the potential importance of CCCs in volume regulation in the central nervous system, but I could not really see the data from the two citations. These are difficult measurements to make in non-cultured cells so we are still far behind on understanding their role in CNS brain water homeostasis (as also stated in Kahle et al., 2015). But if the authors are aware of the data illustrating this statement, please provide the references.

Results:

“1 μ M ZT-1a inhibited GSK3 β activity by $60 \pm 6\%$ compared to DMSO control, representing high kinase selectivity. However, GSK3 β Ser9 phosphorylation was not inhibited by 3 μ M ZT-1a in HEK293 cells.”

I do not fully understand how an inhibition of 60% for another kinase (GSK3) can be interpreted as high selectivity. Please describe.

Fig. 1. It is very confusing that the authors first describe part of figure 1, then describe other issues demonstrated in the supplementary section, and then back to some (partly overlapping) issues in fig. 1 again. Here the authors already refer to panel 1D – but that is the K⁺ transport – not the phosphorylation they describe (which is just summary of the WB described a few paragraphs back?). It is all followed by a new paragraph that also describes part of figure 1, but now it is the functional part in figure 1D, that is described. Can it not be described together? It should be straight forward, but does become very confusing the way it is presented. Figure 1D is still very confusing to me. I do not see a furosemide-sensitive uptake. How do you know that what you look at is KCC-mediated? And why is there virtually no difference as to whether the experiments are carried out in isotonic or hypotonic low Cl⁻ solutions? Should you not have an effect on KCC3-mediated K⁺ transport? Please describe clearly in the result text. And to compare a mutant transfection with a WT transfection, you would need to illustrate that you have the same amount of protein expressed prior to state anything about their activity, would you not? And for supplementary Fig 2B, activation of NKCC1 by hypotonic solutions (with low Cl⁻) is also confusing – when it is generally assumed that cell shrinkage would activate the transporter? I would highly recommend describing the process in the result section so that the reader can follow your train of thought. And if this is the important part to show that ZT-1a inhibits one and activates another, it is important to have both figures in the real text and not in supplementary (in my opinion).

Fig. 2. “Immunoblot of rat choroid plexus lysates demonstrated robust immunoreaction with antibodies targeting KCC1, KCC3 and KCC4, whereas KCC2 abundance was below the limit of detection, consistent with its documented neuronal-specific expression pattern”

As I read the figure, it appears that in CPE lysates there is no KCC staining – only when you perform the pull down with KCC site-2 Ab, do you get bands on the gel. Please describe exactly that in the text – the way you present it makes it look like you have robust KCC expression, which you apparently do not – otherwise you should be able to detect it in the lysate. Only when you up-concentrate by IP, can you detect your protein (so most likely low abundance). This extra step must be apparent to the reader so he/she can evaluate the importance of KCCs in CPe.

It is hard to follow the panels, when Panel 2B is described at the end of the section – after Panel 2C. Like-wise – the WB shows SPAK and NKCC1 at the top and the KCCs (again immunoprecipitated) at the bottom – but the quantifications on the side are in the opposite order. Again, to help the reader follow the story, please arrange your figures to follow the text.

Fig. 3B. "As shown in Figure 3B, hyperosmotic stress led to 45 % cell volume decrease at an initial rate of 24 ± 5 % vol/min."

While you in this set-up probably can evaluate the final volume, the initial rate would probably need a lot more work to achieve accurately – and I would recommend to leave out this number. The majority of the time, the wash-in of the solution is the rate-limiting step – not the swelling. So you need to 1. Illustrate that you can wash-in the solution faster than the cells shrink and 2. Sample at a higher rate. To get a straight line, like you do here, you need more points on the linear part of the cell shrinkage – it looks like you make a straight line between two points – which is strictly speaking not really acceptable. But in short – leave out the initial rate and you should be fine.

Fig. 6. I am sorry for being so confused, but I simply cannot follow the experiments illustrated with those described in the text. The authors first mention phosphorylation of SPAK (over phosphorylation of pOSPR1) – but that is not what is shown in panel B – it is first some phosphorylation sites over KCC3 (two of those) followed by KCC3 over NKA. SPAK is in the last line (panel 7 of 9 and here it is pSer373/SPAK and not what was listed in the text?). It is followed by SPAK/NKA – not mentioned in the text? Then comes NKCC1 – and then KCC3 (opposite to what is in the text and not clear what addresses what – and that you are looking at IP again and not direct quantification. One has the sense that the figures are not described and the text not written to aid the reader in understanding what is going on – it takes much longer to navigate in this manuscript that it ought to. The data are most likely fine, but it is so challenging to really figure that out.

Fig. 7. To deem that ZT-1a is superior, the authors would necessarily need to show some data to illustrate this effect and provide the associated statistics. On figure 7, it is clear that infarct volume and swelling is nicely reduced with Closantel – but I see no numeric comparison to ZT-1a, although the paragraph title promises that? They may survive longer, but that is not the question, since I assume they survive fairly well when you can inhibit the stroke-induced infarct and swelling to that degree? In the discussion, the authors point out that closantel cannot be used in humans due to retinal toxicity. Why not just use that angle instead of stating that it is better based on data that are not shown? Although again in the discussion, it is stated that Zt-1a is more potent than closantel. Did the authors show that anywhere? And again later in the discussion: "ZT-1a is superior to both the anti-parasitic drug Closantel and the pan-WNK inhibitor WNK463 in reducing ischemic brain damage" – I did not see any experiment targeted to compare ZT-1a and closantel – and saw no statistics on such a comparison. How can the authors provide exactly such statement repeatedly with no data to show that? Or did I miss something?

Page 11, line 39: "The CPe secretes higher per-cell volumes of fluid (~500 ml/day CSF) than any other epithelium.". Please provide reference for the CPe being superior to any other epithelium.

Reviewer #3:

Remarks to the Author:

I am happy with the revisions made by Zhang et al. The manuscript is now far stronger and clearer than the original submission.

Dear Editors and Reviewers at Nature Communications,

We thank the reviewers for their critical review of our R1 revision and additional suggestions. We have revised the manuscript accordingly. Following are our point-by-point responses,

Reviewer #1 (Remarks to the Author):

General Comments:

The authors have been very responsive to my comments and suggestions for revision. This is now a much-improved manuscript.

Specific Comments

Q1: I have only one remaining issue. It relates to new data presented in Figure 3 of the revised manuscript. There is an inconsistency between the Results text and the legend regarding the level of hypertonicity used in the experiments. The legend indicates that it was 515 mOsm but the Results text indicates that 370 mOsm was used. Also, the authors state the ZT-1a accelerated the hypertonicity-mediated cell shrinkage. That does not make sense. When cells are exposed to hypertonic solutions they shrink, acting as a perfect (or near perfect) osmometer as water quickly moves out of the cell down its concentration gradient. The resulting cell shrinkage then stimulates NKCC activity to mediate a regulatory volume increase. This needs to be corrected. I have three other concerns about this figure and how it is presented and interpreted. The first is that if the cells were truly switched from 310 mOsm to 515 mOsm the cell volume would be expected to decrease to 58% of control volume. In the figure shown that is approximately what occurred in the control cells but volume decreased further in cells treated with bumetanide or ZT-1a. Do the authors have an explanation for that? Why would cells shrink further when NKCC is inhibited? Second, in the control cells the volume at first decreased rapidly as expected but then continued to decrease at a slower rate. I suspect that this could be an artifact of evaluating cell volume using Calcein. Third, the "volume recovery rate" that the authors refer to does not seem to be a true volume recovery, at least one not mediated by cellular mechanisms because the cells are simply being switched from hypertonic back to isotonic medium. It's important to clarify here what is true volume regulatory mechanism versus passive changes in volume due to altering the osmotic gradient across the cell membrane. Could the authors please address these issues?

Response:

(a). We apologize for the erroneous "515 mOsm" in the Figure 5 legend (Figure 3 legend in R1 version), which has now been corrected to "370 mOsm" on page 15, line 32.

(b). Regulatory volume increase (RVI) and recovery rate measurement: we agree with Reviewer #1 that our averaged data failed to illustrate RVI in cultured primary neurons and fibroblasts, and that our originally described measurement of "volume recovery rate" reflected the cell volume recovery upon switching to the isotonic solution, rather than RVI. We re-analysed our data by measuring each individual cell volume changes. We detected RVI in response to hypertonicity-mediated osmotic shrinkage in ~ 30% of neurons (16/49 cells from three cultures; see representative cell volume change in revised Figure 5B). However, no neurons exhibited RVI in the presence of either NKCC1 inhibitor BMT or SPAK inhibitor ZT-1a (0/46 or 0/49 cells, respectively). We interpreted these data as strongly suggesting that RVI responses are dependent on SPAK-NKCC1 signalling. However, we do not understand why some cells fail to exhibit RVI during 20 min exposure to hypertonic solutions. Variable neuronal volume changes in response to osmotic stress have been reported previously (Andrew et al., *Cerebral Cortex*, 2007, 17 787-802; Murphy et al., *Front Cellular Neurosci.*, 2017, 11). Neuronal culture conditions, cell density, and variable proportions of cocultured astrocytes may each contribute to expression of neuronal RVI, as may cell-by-cell variability in NKCC1 expression. These factors will need testing in future studies, preferably using new approaches with better sensitivity and faster data acquisition (both limitations of the Calcein dye-based method; Model M., *Cytometry*, 2018, 93A, 281-296).

(c). Explanation of severe cell shrinkage in the BMT- or ZT-1a-treated cells in response to hypertonic osmotic stress: we believe that WNK-SPAK-NKCC1 complex and other volume regulatory signalling pathways are activated upon cell shrinkage and immediately counteract cell volume decrease. Thus, when the anti-cell shrinkage mechanism is blocked, cells will display variable cell shrinkage responses. We

repeatedly observed severe cell shrinkage in fibroblasts treated with BMT- or ZT-1a (Figure S9) or in HEK293 epithelial cells when WNK1 was knocked out (Roy et al., *AJP Renal Physiology*, 2015, 308, F366-F376).

(d). Slower cell shrinkage: we detected slower cell shrinkage in neurons subjected to 20 min hypertonic stress in the presence of ZT-1a or BMT, but the underlying mechanisms remain unclear. This slower cell shrinkage did not reflect cell damage-mediated dye loss, because the cells recovered their initial volume upon return to isotonic solutions, and we were able to calibrate with the osmolality standard buffers at the end of the experiment. It's possible that the slower cell shrinkage represents of hypertonicity-activated Na⁺ influx exceeding K⁺ efflux, future studies are needed to explore the possibility.

References:

- Andrew, R.D., Labron, M.W., Boehnke, S.E., Carnduff, L., and Kirov, S.A. (2007). Physiological evidence that pyramidal neurons lack functional water channels. *Cereb Cortex* 17, 787-802.
- Model, M.A. (2018). Methods for cell volume measurement. *Cytometry A* 93, 281-296.
- Murphy, T.R., Davila, D., Cuvelier, N., Young, L.R., Lauderdale, K., Binder, D.K., and Fiocco, T.A. (2017). Hippocampal and Cortical Pyramidal Neurons Swell in Parallel with Astrocytes during Acute Hypoosmolar Stress. *Front Cell Neurosci* 11, 275.
- Roy, A., Goodman, J.H., Begum, G., Donnelly, B.F., Pittman, G., Weinman, E.J., Sun, D., and Subramanya, A.R. (2015). Generation of WNK1 knockout cell lines by CRISPR/Cas-mediated genome editing. *Am J Physiol Renal Physiol* 308, F366-376.

Reviewer #2 (Remarks to the Author):

General Comments:

This revised manuscript by Zhang et al describes a novel inhibitor targeting the SPAK kinase and subsequent phosphorylation of NKCC1/KCCs. The manuscript addresses inclusion of this inhibitor in settings of CSF hypersecretion and brain edema formation. The revised manuscript is interesting and has addressed a few of the critique points raised in the last round. I, however, still find it difficult to navigate in some of the figures and the associated (brief) text, which are sometimes not provided in the same order. I still have some concerns with the manuscript as it stands and I (again) would urge the authors to consider re-writing parts of the manuscript to clarify some points and aid the reader in understanding the study.

We thank Reviewer 2 for the constructive comments that we have systematically addressed below.

Specific Comments

Q2: Introduction: "Vectorial ion transport across apical and basolateral membranes of the choroid plexus epithelium (CPE), accompanied by osmotically obligated transport of water, results in daily cerebrospinal fluid (CSF) secretion of > 500 cc/day into brain ventricular spaces¹".

The particular study cited argued exactly against the point of osmotically obliged transport of water, so puzzling that the authors would cite that in this particular context?

Response: "Vectorial ion transport across apical and basolateral membranes of the choroid plexus epithelium (CPE), accompanied by transport of water, cotransported (1-3) or osmotically obliged (4-7), results in daily cerebrospinal fluid (CSF) secretion of > 500 cc/day into brain ventricular spaces."

1. Steffensen AB, et al. Cotransporter-mediated water transport underlying cerebrospinal fluid formation. *Nat Commun* 9, 2167 (2018).
2. Jin SC, et al. SLC12A ion transporter mutations in sporadic and familial human congenital hydrocephalus. *Mol Genet Genomic Med*, e892 (2019).
3. Karimy JK, et al. Inflammation-dependent cerebrospinal fluid hypersecretion by the choroid plexus epithelium in posthemorrhagic hydrocephalus. *Nature medicine* 23, 997-1003 (2017).

4. Wald A, Hochwald GM, Malhan C. The effects of ventricular fluid osmolality on bulk flow of nascent fluid into the cerebral ventricles of cats. *Experimental brain research* 25, 157-167 (1976).
5. Krishnamurthy S, Li J, Schultz L, Jenrow KA. Increased CSF osmolarity reversibly induces hydrocephalus in the normal rat brain. *Fluids Barriers CNS* 9, 13 (2012).
6. Wald A, Hochwald GM, Gandhi M. Evidence for the movement of fluid, macromolecules and ions from the brain extracellular space to the CSF. *Brain research* 151, 283-290 (1978).
7. Klarica M, Mise B, Vlastic A, Rados M, Oreskovic D. "Compensated hyperosmolarity" of cerebrospinal fluid and the development of hydrocephalus. *Neuroscience* 248, 278-289 (2013).

Q3: "Impaired ionic homeostasis in CPe can result in hydrocephalus (accumulation of excess CSF in the brain ventricles), as in the settings of intraventricular hemorrhage (IVH) and infection^{2, 3}"

Has it ever been shown that it is the ion homeostasis in the CP epithelial cells that result in hydrocephalus? I looked through one of the cited articles but was unable to find that statement supported there? Please verify and/or rephrase – or cite the study by Karimy et al., which illustrates hyperactivity of an ion transporter as the etiology.

Response: Our recent findings identify the *KCC3* and *KCC4* genes as associated with congenital hydrocephalus (CH), and implicate genetically encoded impairments in ion transport for the first time in CH pathogenesis (see ref: Jin SC, et al. SLC12A ion transporter mutations in sporadic and familial human congenital hydrocephalus. *Mol Genet Genomic Med*, e892 (2019).).

Therefore we added this reference in "Impaired ionic homeostasis in CPe can result in hydrocephalus (accumulation of excess CSF in the brain ventricles), as in the settings of intraventricular hemorrhage (IVH) and infection^{1, 2, 3}."

1. Steffensen AB, et al. Cotransporter-mediated water transport underlying cerebrospinal fluid formation. *Nat Commun* 9, 2167 (2018).
2. Jin SC, et al. SLC12A ion transporter mutations in sporadic and familial human congenital hydrocephalus. *Mol Genet Genomic Med*, e892 (2019).
3. Karimy JK, et al. Inflammation-dependent cerebrospinal fluid hypersecretion by the choroid plexus epithelium in posthemorrhagic hydrocephalus. *Nature medicine* 23, 997-1003 (2017).

Q4: Again (see Q23 from last round): "These evolutionarily-conserved transporters²³, are of particular importance in regulation of ion and water homeostasis in mammalian central nervous system (CNS)^{24, 25}. The coordinated regulation of CCC function is important for cell volume regulation in most brain cells, preventing excessive cell swelling or shrinkage in response to osmotic or ischemic stress^{22, 26}."

I certainly do not disagree on the potential importance of CCCs in volume regulation in the central nervous system, but I could not really see the data from the two citations. These are difficult measurements to make in non-cultured cells so we are still far behind on understanding their role in CNS brain water homeostasis (as also stated in Kahle et al., 2015). But if the authors are aware of the data illustrating this statement, please provide the references.

Response: We have provided some relevant references to illustrate this statement in "These evolutionarily-conserved transporters²⁷, are of particular importance in regulation of ion and water homeostasis in mammalian central nervous system (CNS)^{28, 29}. The coordinated regulation of CCC function is important for cell volume regulation in most brain cells, preventing excessive cell swelling or shrinkage in response to osmotic or ischemic stress^{26, 30, 31, 32, 33, 34}."

26. Zhang J, et al. Functional kinomics establishes a critical node of volume-sensitive cation-Cl⁻ cotransporter regulation in the mammalian brain. *Scientific reports* 6, 35986 (2016).

30. Kahle KT, et al. K-Cl cotransporters, cell volume homeostasis, and neurological disease. *Trends Mol Med* 21, 513-523 (2015).
31. Flores B, Schornak CC, Delpire E. A role for KCC3 in maintaining cell volume of peripheral nerve fibers. *Neurochem Int* 123, 114-124 (2019).
32. Jayakumar AR, Panickar KS, Curtis KM, Tong XY, Moriyama M, Norenberg MD. Na-K-Cl cotransporter-1 in the mechanism of cell swelling in cultured astrocytes after fluid percussion injury. *Journal of neurochemistry* 117, 437-448 (2011).
33. MacVicar BA, Feighan D, Brown A, Ransom B. Intrinsic optical signals in the rat optic nerve: role for K(+) uptake via NKCC1 and swelling of astrocytes. *Glia* 37, 114-123 (2002).
34. Jayakumar AR, et al. Na-K-Cl Cotransporter-1 in the mechanism of ammonia-induced astrocyte swelling. *The Journal of biological chemistry* 283, 33874-33882 (2008).

Q5: “1 μ M ZT-1a inhibited GSK3 β activity by $60 \pm 6\%$ compared to DMSO control, representing high kinase selectivity. However, GSK3 β Ser9 phosphorylation was not inhibited by 3 μ M ZT-1a in HEK293 cells.”

I do not fully understand how an inhibition of 60% for another kinase (GSK3) can be interpreted as high selectivity. Please describe.

Response: We assessed the kinase selectivity of ZT-1a against a broad panel of 140 recombinant kinases. At a concentration of 1 μ M, ZT-1a exhibited >50% inhibition of only a single kinase among the 140 tested, GSK3 β ($60 \pm 6\%$). However, at 3 μ M ZT-1a [the concentration we used in cells to induce dephosphorylation of SPAK (Ser373), NKCC1(Thr203/207/212) and KCC (site 1 and site 2)], we found that GSK3 β Ser9 phosphorylation was inhibited neither in HEK293 cells (Figure S4A-B), nor in ZT-1a-treated ischemic brain. We are thus careful to state that ZT-1a is not a mono-selective inhibitor of SPAK. In light of the Reviewer's comments, we refer to ZT-1a as a 'selective' inhibitor rather than 'highly selective'. For reference, the highly selective Her2 inhibitor lapatinib inhibited 12 kinases with a dissociation constant ($K_d < 3 \mu$ M) against a panel of 317 kinases assayed (12/317), and imatinib inhibited 26 kinases with $K_d < 3 \mu$ M (26/317) (Ref: Karaman, M.W. et al. *Nat. Biotechnol.* 26, 127-32 (2008)).

Q6: Fig. 1. It is very confusing that the authors first describe part of figure 1, then describe other issues demonstrated in the supplementary section, and then back to some (partly overlapping) issues in fig. 1 again. Here the authors already refer to panel 1D – but that is the K⁺ transport – not the phosphorylation they describe (which is just summary of the WB described a few paragraphs back?). It is all followed by a new paragraph that also describes part of figure 1, but now it is the functional part in figure 1D, that is described. Can it not be described together? It should be straight forward, but does become very confusing the way it is presented.

Figure 1D is still very confusing to me. I do not see a furosemide-sensitive uptake. How do you know that what you look at is KCC-mediated? And why is there virtually no difference as to whether the experiments are carried out in isotonic or hypotonic low Cl solutions? Should you not have an effect on KCC3-mediated K⁺ transport? Please describe clearly in the result text. And to compare a mutant transfection with a WT transfection, you would need to illustrate that you have the same amount of protein expressed prior to state anything about their activity, would you not? And for supplementary Fig 2B, activation of NKCC1 by hypotonic solutions (with low Cl⁻) is also confusing – when it is generally assumed that cell shrinkage would activate the transporter? I would highly recommend describing the process in the result section so that the reader can follow your train of thought. And if this is the important part to show that ZT-1a inhibits one and activates another, it is important to have both figures in the real text and not in supplementary (in my opinion).

Response: to increase clarity, we have divided 'Figure 1' into the following three figures:

Figure 1 (contains Figure 1A in R1 version). Design a novel SPAK-CCC modulator, ZT-1a, through a hybrid design strategy for new WNK pathway inhibitors.

Figure 2 (contains Figures 2B, 2C and S2A in R1 version). The SPAK-CCC modulator ZT-1a potently suppresses phosphorylation of KCC3A and NKCC1. In this figure, we have just enough space to include all the immunoblot data and quantitation for ZT-1a, Closantel, STOCK1S-14279 and STOCK1S-50699.

Figure 3 (contains Figures 2D and S2B in R1 version). Inhibition of SPAK/OSR1 kinase activity by ZT-1a correlates with transport activity of KCC3A and NKCC1.

We repeated these experiments, and now include a furosemide-sensitive uptake assay as described in “Kahle KT et al. Peripheral motor neuropathy is associated with defective kinase regulation of the KCC3 cotransporter. *Sci Signal*. 2016 Aug 2;9(439):ra77”. The furosemide (Furo)-treated cells, either transfected with empty vector or KCC3 wt cDNA, have significantly decreased K⁺ flux ($p < 0.01$; $n = 3$; Figure 3A), in isotonic or hypotonic low Cl⁻ conditions. Compared with isotonic Cl⁻ conditions, lower KCC3 activity was consistently observed in wild-type (WT) HEK293 cells, with maximal KCC3 phosphorylation at Thr991/1048 in hypotonic low [Cl⁻] conditions. The observed higher NKCC1 activity was as expected, given that NKCC1 is maximally phosphorylated following hypotonic low Cl⁻ conditions,. The Bumetanide (Bum)-treated cells, either transfected with empty vector or NKCC1 wt cDNA, have significantly decreased K⁺ flux in isotonic or in hypotonic low Cl⁻ conditions ($p < 0.001$; $n = 3$; Figure 3B).

Q7: Fig. 2. “Immunoblot of rat choroid plexus lysates demonstrated robust immunoreaction with antibodies targeting KCC1, KCC3 and KCC4, whereas KCC2 abundance was below the limit of detection, consistent with its documented neuronal-specific expression pattern”

As I read the figure, it appears that in CPE lysates there is no KCC staining – only when you perform the pull down with KCC site-2 Ab, do you get bands on the gel. Please describe exactly that in the text – the way you present it makes it look like you have robust KCC expression, which you apparently do not – otherwise you should be able to detect it in the lysate. Only when you up-concentrate by IP, can you detect your protein (so most likely low abundance). This extra step must be apparent to the reader so he/she can evaluate the importance of KCCs in CPe.

It is hard to follow the panels, when Panel 2B is described at the end of the section – after Panel 2C. Likewise – the WB shows SPAK and NKCC1 at the top and the KCCs (again immunoprecipitated) at the bottom – but the quantifications on the side are in the opposite order. Again, to help the reader follow the story, please arrange your figures to follow the text.

Response: For Figure 4 (Figure 2 in R1 version), we have tried to describe exactly this in the text: “Although immunospecific, none of these antibodies detected endogenous KCCs from rat CPe lysates through direct immunoblot. However, immunoblot of immunoprecipitated fractions demonstrated robust immunoreaction with antibodies targeting KCC1, KCC3 and KCC4, whereas CPe KCC2 abundance remained below the detection limit, consistent with its documented neuron-specific expression pattern.”

We have clarified Figure 4 by rearranging Figure 4B and Figure 4C.

Q8: Fig. 3B. “As shown in Figure 3B, hyperosmotic stress led to 45 % cell volume decrease at an initial rate of 24 ± 5 % vol/min.”.

While you in this set-up probably can evaluate the final volume, the initial rate would probably need a lot more work to achieve accurately – and I would recommend to leave out this number. The majority of the time, the wash-in of the solution is the rate-limiting step – not the swelling. So you need to 1. Illustrate that you can wash-in the solution faster than the cells shrink and 2. Sample at a higher rate. To get a straight line, like you do here, you need more points on the linear part of the cell shrinkage – it looks like you make a straight line between two points – which is strictly speaking not really acceptable. But in short – leave out the initial rate and you should be fine.

Response: We agree with Reviewer # 2 that the resolution of our current method is limited by its sample collection rate and rate of solution change. Therefore, we removed presentation of the initial rate of cell volume decrease on (page 8, line 2-5) and in the revised Figure 5.

Q9: Fig. 6. I am sorry for being so confused, but I simply cannot follow the experiments illustrated with those described in the text. The authors first mention phosphorylation of SPAK (over phosphorylation of pOSPR1) – but that is not what is shown in panel B – it is first some phosphorylation sites over KCC3 (two of those) followed by KCC3 over NKA. SPAK is in the last line (panel 7 of 9 and here it is pSer373/SPAK and not what was listed in the text?). It is followed by SPAK/NKA – not mentioned in the text? Then comes NKCC1 – and then KCC3 (opposite to what is in the text and not clear what addresses what – and that you are looking at IP again and not direct quantification. One has the sense that the figures are not described and the text not written to aid the reader in understanding what is going on – it takes much longer to navigate in this manuscript that it ought to. The data are most likely fine, but it is so challenging to really figure that out.

Response: We have rearranged the figure to clarify it:

'We therefore examined ZT-1a effects on phosphorylation of SPAK/OSR1, NKCC1, and KCC3 in ischemic mouse brains. Ischemic stroke increased phosphorylation of pSPAK (pSer373)/pOSR1 (pSer325) by ~1.5 fold ($p < 0.05$, $n = 5$), phosphorylation of pNKCC1 (pThr203/207/212) by ~1.6 fold ($p < 0.05$, $n = 5$), and phosphorylation of pKCC3 pThr991 ($p < 0.05$, $n = 5$) and pThr1048 by ~ 1.3-fold ($p < 0.05$) in membrane protein fractions from the ipsilateral (IL) cortical hemisphere at 24 hr reperfusion in vehicle-control treated mice, without significant change in corresponding total protein levels (Figures 8A-B). Post-stroke administration of ZT-1a in mice prevented ischemia-induced increases of pSPAK/pOSR1, pNKCC1, and pKCC3 ($p < 0.05$, $n = 5$) without affecting corresponding total protein expression (Figures 8A-B; Figure S11). These results indicate that ZT-1a inhibits the SPAK-dependent up-regulation of NKCC1 and KCC3 phosphorylation in ischemic brains.'

Q10: Fig. 7. To deem that ZT-1a is superior, the authors would necessarily need to show some data to illustrate this effect and provide the associated statistics. On figure 7, it is clear that infarct volume and swelling is nicely reduced with Closantel – but I see no numeric comparison to ZT-1a, although the paragraph title promises that? They may survive longer, but that is not the question, since I assume they survive fairly well when you can inhibit the stroke-induced infarct and swelling to that degree? In the discussion, the authors point out that closantel cannot be used in humans due to retinal toxicity. Why not just use that angle instead of stating that it is better based on data that are not shown? Although again in the discussion, it is stated that Zt-1a is more potent than closantel. Did the authors show that anywhere? And again later in the discussion: "ZT-1a is superior to both the anti-parasitic drug Closantel and the pan-WNK inhibitor WNK463 in reducing ischemic brain damage" – I did not see any experiment targeted to compare ZT-1a and closantel – and saw no statistics on such a comparison. How can the authors provide exactly such statement repeatedly with no data to show that? Or did I miss something?

Response: As suggested by Reviewer #2, we have softened our statements by removing the term "superior" from the paragraph title and the results and revised the following passages:

On page 9, line 40-41: "Closantel shows neuroprotection but WNK463 fails to reduce ischemic brain injury in mice"

On page 10, line 5-6: "These data demonstrate ZT-1a and Closantel are effective for *in vivo* inhibition of the WNK-SPAK-CCC pathway in the mouse model of ischemic brain injury".

On page 11, line 43-46: "Although, ZT-1a and the anti-parasitic drug Closantel exhibit similar effects in reducing ischemic brain damage, Closantel has shown toxicity in mouse and human^{72, 73}. In this study, we did not test the effect of ZT-1a on the visual system. Additional pharmacokinetics and toxicity studies of ZT-1a, especially on retina, warrant further investigation for future clinical translation."

72. Tabatabaei SA, *et al.* Closantel; a veterinary drug with potential severe morbidity in humans. *BMC Ophthalmol* **16**, 207 (2016).

73. Essabar L, Meskini T, Ettair S, Erreimi N, Mouane N. Harmful use of veterinary drugs: blindness following Closantel poisoning in a 5-year-old girl. *Asia Pacific Journal of Medical Toxicology* **3**, 173–175 (2014).

Q11: Page 11, line 39: “The CPe secretes higher per-cell volumes of fluid (~500 ml/day CSF) than any other epithelium.”. Please provide reference for the CPe being superior to any other epithelium.

Response: We have provided new reference 31 in support of this statement:

Dankier HH, Brown PD, Praetorius J. Cerebrospinal fluid secretion by the choroid plexus. *Physiological reviews* **93**, 1847-1892 (2013).”

Reviewer #3 (Remarks to the Author):

I am happy with the revisions made by Zhang et al. The manuscript is now far stronger and clearer than the original submission.

Response: Many thanks!

Reviewers' Comments:

Reviewer #1:

Remarks to the Author:

The authors have responded to my remaining concerns. I have no further comments.

Reviewer #2:

Remarks to the Author:

The authors have satisfactorily replied to the raised critique points. Some minor issues are that cultured astrocytes appear to act quite differently with NKCC1-dependent volume regulation than in vivo/brain slices (so some references may not truly reflect in vivo situations), but that is of minor importance given that the study is not focused on this.

Please provide the animal number for CSF secretion (Fig. 4C), this is not provided in results or in figure legend.

Dear Editors and Reviewers at Nature Communications,

We thank the reviewers for their critical review of our R2 revision and additional suggestions. We have revised the manuscript accordingly, using the 'track changes' feature in Word. Following are our point-by-point responses.

Reviewer #1 (Remarks to the Author):

General Comments:

The authors have responded to my remaining concerns. I have no further comments.

Reviewer #2 (Remarks to the Author):

General Comments:

Q1: The authors have satisfactorily replied to the raised critique points. Some minor issues are that cultured astrocytes appear to act quite differently with NKCC1-dependent volume regulation than in vivo/brain slices (so some references may not truly reflect in vivo situations), but that is of minor importance given that the study is not focused on this.

Response: we agree that some references may not truly reflect in vivo situations.

Q2: Please provide the animal number for CSF secretion (Fig. 4C), this is not provided in results or in figure legend.

Response: we have provided n=3, three independent experiments, for all group, in the figure legend of Figure 4C.